# GATA2 regulates mast cell identity and responsiveness to antigenic stimulation by promoting chromatin remodeling at super-enhancers

Yapeng Li[1], Junfeng Gao [1], Mohammad Kamran[1], Laura Harmacek[2], Thomas Danhorn [2], Sonia M. Leach[1,2], Brian P. O'Connor[2], James R. Hagman [1,3] & Hua Huang [1,3 ✉]

Mast cells are critical effectors of allergic inflammation and protection against parasitic infections. We previously demonstrated that transcription factors GATA2 and MITF are the mast cell lineage-determining factors. However, it is unclear whether these lineage-determining factors regulate chromatin accessibility at mast cell enhancer regions. In this study, we demonstrate that GATA2 promotes chromatin accessibility at the super-enhancers of mast cell identity genes and primes both typical and super-enhancers at genes that respond to antigenic stimulation. We find that the number and densities of GATA2- but not MITF-bound sites at the super-enhancers are several folds higher than that at the typical enhancers. Our studies reveal that GATA2 promotes robust gene transcription to maintain mast cell identity and respond to antigenic stimulation by binding to super-enhancer regions with dense GATA2 binding sites available at key mast cell genes.

[1] Department of Immunology and Genomic Medicine, National Jewish Health, Denver, CO 80206, USA. [2] Center for Genes, Environment and Health, National Jewish Health, Denver, CO 80206, USA. [3] Department of Immunology and Microbiology, University of Colorado Anschutz Medical Campus, Aurora, CO 80045, USA. ✉email: huangh@njhealth.org

Mast cells (MCs) are critical effectors in immunity that originally evolved to protect against parasitic infections. In modern living conditions, MCs are much more frequently associated with allergic inflammation[1,2]. MCs reside in the skin, mucosal linings of the lung and gut, and connective tissues surrounding blood vessels. In these tissue contexts MCs are among the first cells to encounter antigens, which can be captured by IgE antibodies. MCs express the high-affinity receptor for IgE (FcεRI). Aggregation of FcεRI bound with IgE and their cognate antigens leads to MC activation and rapid degranulation to release pre-existing inflammatory mediators including histamine, heparin, and proteases. MC activation also directs de novo gene transcription of additional inflammatory mediators, including cytokine and chemokine genes.

Transcription factors (TFs) GATA binding protein 2 (GATA2) and Microphthalmia-associated transcription factor (MITF) play critical roles in the differentiation of MC progenitor cells into the MC lineage[3,4]. GATA2 is critical for survival and proliferation of hematopoietic stem cells[4,5], granulocyte-monocyte progenitor differentiation[6], and basophil and MC differentiation[4,7]. Spontaneous mutations in the *Mitf* gene lead to the failure of MC progenitor cells to differentiate into mature MCs[3,8,9]. GATA2 is also essential in maintaining the MC identity once MCs are fully committed to the MC lineage. We and others demonstrated that MC-specific deletion of the *Gata2* gene results in the failure of MCs to maintain the MC identity[10,11]. We have demonstrated that the *Mitf* gene is highly expressed in MCs but not in basophils[12] and that overexpression of the *Mitf* gene is sufficient to drive the differentiation of pre-BMPs into MCs[12]. Together, this evidence supports a model in which GATA2 and MITF are lineage-determining TFs (LDTFs) in MCs and GATA2 is required for MC identity maintenance.

Little is known concerning how GATA2 and MITF regulate target gene transcription in MCs. Notably, enhancers that drive MC-specific transcription have not been localized. Generally, enhancers are regulatory modules of a few hundred base pairs in length located within genes or in intergenic regions. They typically comprise clusters of TF-binding sites that bind sequence-specific DNA-binding TFs and associated factors[13]. Enhancers activate gene transcription by mediating the assembly of higher-order functional domains with promoters[14]. Enhancers also provide binding hubs for signal-dependent TFs (SDTFs) that respond to the stimulation of receptors by external ligands. Together, signals triggered by ligand-bound receptors modulate activities of enhancers, which in turn drive gene transcription necessary for cell development and function[15].

Super-enhancers differ from typical enhancers in multiple ways[16,17]. Super-enhancers comprise larger tracts of genomic DNA that include multiple smaller constituent enhancers. Super-enhancers are often associated with genes that confer cell identities[17,18] and are enriched in genetic variants that may contribute to disease progression and severity. Originally, the concept of super-enhancers was developed in embryonic stem cells[17]. Since then, super-enhancers have been described in T cells[19], B cells[20,21] and macrophages[22] as well as in many other cell systems[23]. However, it has not been determined whether super-enhancers play similar pivotal roles in regulating transcription and cell identity in MCs.

Remarkable progress has been made in identifying putative enhancers genome-wide. Epigenomic studies have demonstrated that monomethylation of lysine residue 4 on histone 3 (H3K4me1) marks genes that are poised to be transcribed, while acetylation of lysine residue 27 on histone 3 (H3K27ac) identifies genes that are actively being transcribed[24–27]. In this study, we used histone modifications to identify typical and super-enhancers in MCs. We found that super-enhancers are enriched

at key MC genes. The number and density of GATA2 bound sites were higher at these super-enhancers relative to typical enhancers. We found that the super-enhancers had higher scores predicting their regulatory potential versus typical enhancers. The key MCs genes with higher regulatory potential scores express much higher levels of mRNA than genes with typical enhancers. We demonstrate that GATA2 promotes chromatin accessibility, which correlates with H3K4me1 and H3K27ac modifications at the super-enhancers of key MC genes. Our studies reveal that the increased frequency of GATA2 binding sites within super-enhancers at key MC genes promotes robust gene transcription that maintains MC identity and the response to antigenic stimulation.

## Results

**Super-enhancers are enriched in the mast cell identity genes.** To identify enhancers of MC lineage genes, we performed H3K4me1 and H3K27ac ChIP-seq on two biological replicates of resting bone marrow-derived MCs (BMMCs). We defined potential enhancers as regions of chromatin marked by H3K4me1 together with H3K27ac modifications. We identified an average total of 20801 potential enhancers (Supplementary Data 1). Super-enhancers are essential for the regulation of genes that control cell differentiation[17]. Using the software Ranking Of Super Enhancer (ROSE)[17], we localized super-enhancers as regions of chromatin that were at least 12.5 kb long and associated with significant H3K27ac modification. We identified an average of total 9,517 typical enhancers and 667 super-enhancers found in two replicates (Fig. 1a, Supplementary Data 2). The average length of the super-enhancers was 13-fold greater than the typical enhancers (Fig. 1a). The genomic distributions of the typical and super-enhancers were quite different. Typical enhancers were mostly located in distal intergenic regions (36.5%), whereas the majority of super-enhancers were localized near promoters (55.8%) and overlapped with gene body exons and introns (16.2%) (Fig. 1b). The genomic locations of the super-enhancers suggest that the close proximity of super-enhancers to promoters makes their interplay more efficient, although many enhancers occur at distances up to several megabases from their interacting promoters[28]. Representative tracks from one of the two biological replicates for the typical and super-enhancers are shown in Fig. 1c. Reproducibility of our NGS sequencing data generated from biological replicates is documented in Supplementary Figs. 1 and 2.

MC identity (ID) genes have been defined by the Immunological Genome Project Consortium as those that are expressed at higher or the same levels in MCs relative to basophils, but at low levels or insignificantly expressed in other blood cell types including T cells, B cells, eosinophils, macrophages, dendritic cells and neutrophils[29]. We compiled a list of 230 MC ID genes from a dataset published by our group[12], merged with the dataset published by the Immunological Genome Project Consortium[29]. In the list of 230 MC ID genes, we also included genes that are not necessarily MC-specific or -enhanced but play critical roles in MC development and function including GATA2 and LYN. We denoted the 230 genes (Supplementary Data 3) as MC ID genes. We defined the other genes expressed in MCs (but not included in the list of MC ID genes) as the non-ID genes (Supplementary Data 3). We found that 36% of the ID genes were associated with super-enhancers, as compared with 4.3% of non-ID genes (Fig. 1d). This frequency represents a striking 8-fold enrichment of super-enhancers at these genes. Because of the reported association of super-enhancers with lineage determining genes, we refer to the 84 ID genes with super-enhancers as the key ID genes (Supplementary Data 3). Examples of the key ID genes

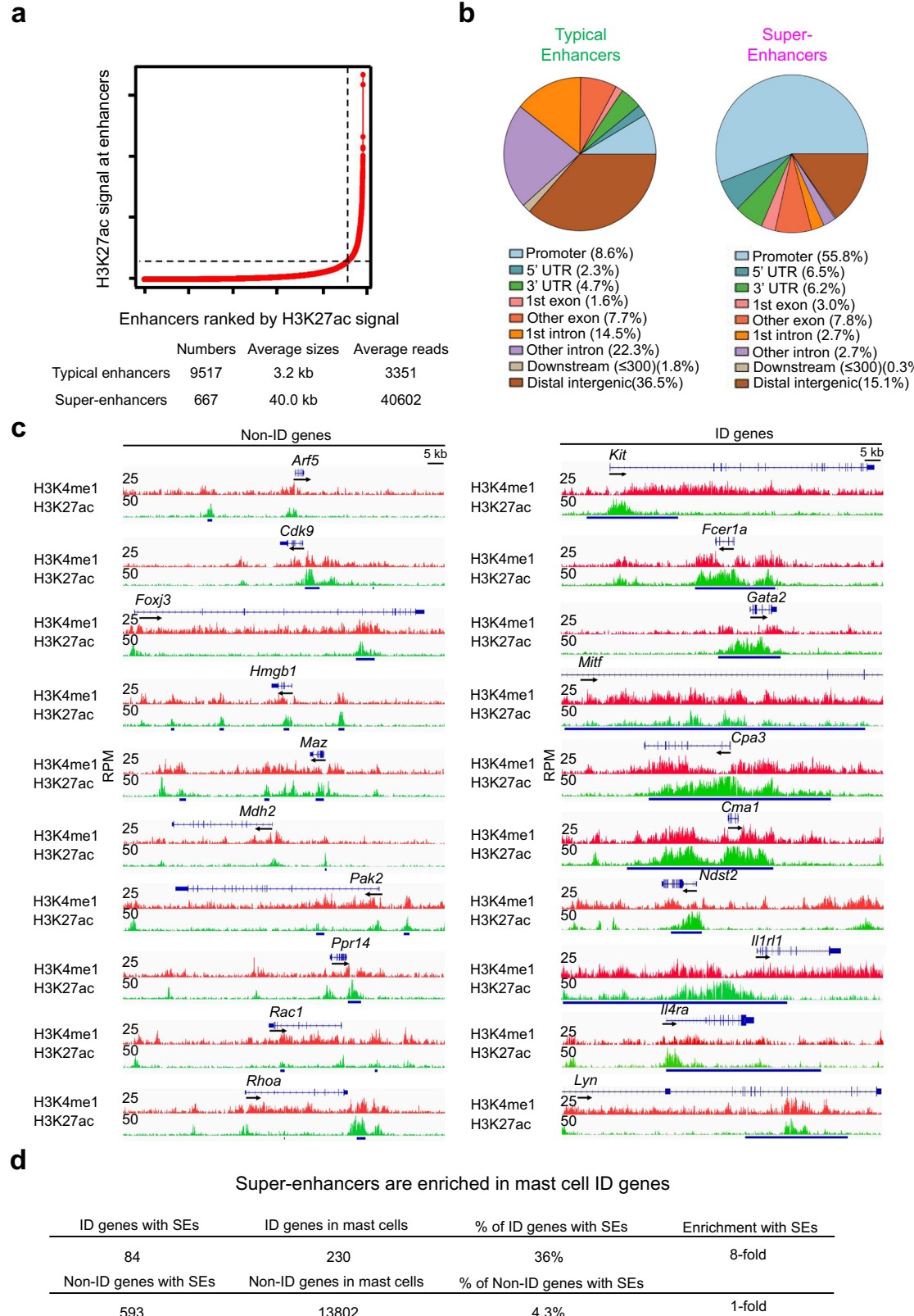

**Fig. 1 Super-enhancers are enriched in mast cell identity genes. a** The H3K27ac ChIP-seq peaks in resting BMMCs were used for analyzing super-enhancers (SEs) and typical enhancers (TEs) using Ranking Of Super Enhancer (ROSE). **b** Genomic distributions of the TEs and SEs. **c** Representative IGV tracks of the mast cell identity (ID) genes and non-ID genes. The small bars below the tracks indicate TEs and the large bars indicate SEs. Data **a–c** represent two biological samples. **d** SEs are enriched in the mast cell ID genes. The ChIP-seq IGV tracks are generated from one biological sample, representing two biological replicates with similar patterns.

include genes that are important for MC functions, including MC surface receptors (*Kit, Fcer1a, Il1rl1 and Il4ra*), transcription factors (*Gata2* and *Mitf*), proteases (*Cpa3, Cma1* and *Mcpt1*), enzymes for sulfur metabolism (*Ndst2, Hs3st1, Hs6st1*) and signaling molecules (*Prkcb* and *Lyn*). These results suggest that super-enhancers direct the programming of key ID genes necessary for MC differentiation and function.

BMMCs represent less mature MCs. Phenotypically, BMMCs resemble mucosal MCs[30,31]. For example, the *Mcpt1* and *Mcpt2* genes were expressed at higher levels in mucosal MCs relative to connective tissue MCs, whereas the *Mcpt5* gene was expressed at higher levels in connective tissue MCs compared to mucosal MCs. In the Immunological Genome Project, Dwyer et al. performed microarray analysis of primary mouse connective tissue MCs isolated from trachea, tongue, esophagus, skin and peritoneum and defined ID genes in these connective tissue MCs. Despite the differences that exist between the *directly* ex vivo MCs and in vitro cultured BMMCs and the difference that exist between the mucosal and connective tissue MCs, our key ID gene list shares 50% of identities with the Dwyer ID gene list (Supplementary Data 4).

**The GATA2 binding motif and occupancy are enriched at the accessible regions within the super-enhancers of the key MC identity genes.** Super-enhancers comprise multiple constituent enhancers that contain clusters of TF binding motifs[17]. To identify TF binding sites associated with super-enhancers, we performed an improved Omni-ATAC-seq protocol that dramatically reduces mitochondria DNA contamination[32] on resting BMMCs ($n = 2$). The majority (63%) of ATAC-seq peaks co-localized within the typical enhancers and super-enhancers. Representative tracks from one biological replicate of the non-ID genes and key ID genes are shown in Fig. 2a. We conducted TF binding motif analysis on the accessible regions within the typical and super-enhancer regions to focus on TF binding motifs that promote gene transcription. The accessible regions that were located outside of the typical and super-enhancer regions could contain TF binding motifs that repress gene transcription and are included in Supplementary Data 5. The possible target genes regulated by these potential repressors are also included in the Supplementary Data 5.

TF binding motifs enriched in these regions included SPDEF, SNAI2, NR2C2, PRDM1 and MEIS2 (Supplementary Fig. 3, left panel), the majority of which have been demonstrated to have transcriptional repressive activities[33–37]. We found that a set of TF binding motifs including those of EGR2, PU.1, GATA2, EGR1 and RUNX1 were enriched at the accessible regions within both the typical enhancers of the non-ID genes and the super-enhancers of the key ID genes (Fig. 2b). Enrichments of these motifs were restricted to accessible regions, typical enhancers, and super-enhancers with significant H3K27ac modification (Fig. 2b). Remarkably, the same set of GATA2 and co-TF binding motifs were enriched in the GATA2 ChIP-seq peaks generated from resting BMMCs available in the public database[38] (Fig. 2b). Among the TFs with enriched TF binding motifs, GATA2, PU.1 and EGR2 have been reported to regulate MC differentiation and functions[4,10,11,39,40], demonstrating that our combined strategy for identifying TF binding motifs at MC enhancers was efficacious.

The enrichment with the same set of GATA2 and co-TF binding motifs at both the typical enhancers and super-enhancers prompted us to investigate whether the relative frequencies of GATA2 binding motifs are different between these two types of enhancers. The frequencies of GATA2 binding motifs were 3-fold higher at the super-enhancers than at the typical enhancers

(Fig. 2c). In contrast, the frequencies of MITF binding motifs were comparable between the super-enhancers and the typical enhancers. To verify that GATA2 binds directly to the accessible regions within the enhancers, we aligned the published GATA2 and MITF ChIP-seq datasets[38] (one biological sample) with our ChIP-seq and ATAC-seq data (Fig. 2d). We calculated the number of GATA2-bound sites and reads on a per gene basis to assess the total frequencies and intensities of GATA2 binding. Averages of GATA2-bound sites and reads per gene were 5.5-fold and 39-fold, respectively, higher at the super-enhancers than at the typical enhancers (Fig. 2e). Because the super-enhancers were much larger than the typical enhancers, we normalized the number of GATA2-bound sites and reads per kilobase per enhancer. Normalized GATA2-bound sites and reads per enhancer were 1.9-fold and 1.3-fold, respectively, higher at the super-enhancers versus typical enhancers (Fig. 2e). In contrast, frequencies and densities of MITF-bound sites at the super-enhancers were similar to those at the typical enhancers (Fig. 2e).

Next, we assessed regulatory potential scores for the enhancers. Tang and colleagues developed an algorithm that utilized a large number of TF ChIP-seq, histone mark ChIP-seq and RNA-seq datasets generated from normal and cancer cell lines[41]. The algorithm estimates regulatory potential scores of enhancers based on the sum of TF factor binding sites as well as the distances of the TF binding sites from transcription start sites (TSS). TF-bound sites that are close to TSS were given higher scores than those that were further away from TSS[42]. Regulatory potential scores calculated using GATA2-bound sites was 2.1-fold higher for the super-enhancers than for the typical enhancers (Fig. 2f). The key ID genes that had higher regulatory potential scores expressed 13.8-fold higher levels of mRNA relative to the non-ID gene that had lower regulatory scores (Fig. 2f). These results suggest that GATA2 promotes robust gene transcription in the key ID genes in MCs by binding to more abundant sites at the accessible regions of super-enhancers.

**GATA2 promotes chromatin accessibility and H3K4me1 and H3K27ac modifications of histones associated with the MC genes.** We hypothesized that binding of GATA2 to typical enhancers and super-enhancers renders the GATA2-bound regions accessible. The bound GATA2 and co-TFs then recruit histone methyltransferase MLL3/4 and histone acetyltransferase P300 to modify histones in the proximity of the bound sites. To test this hypothesis, we deleted the *Gata2* genes in MCs by incubating inducible *Gata2* knockout (*Gata2*[f/f]*Rosa*[Yfp/Yfp]*Tg-CreErt2*[hemi]) BMMCs with 4-hydroxytamoxifen (4-OHT) for five days. The *Gata2* inducible knockout mouse was on the C57BL/6 genetic background, which is different from the genetic background of BALB/c mice that were used to generate the H3K4me1 and H3K27ac ChIP-seq data presented in Figs. 1 and 2. Therefore, we compared gene profiles in C57BL/6 BMMCs and BALB/c BMMCs under resting and stimulated conditions. We found remarkable similarity of gene expression profiles in C57BL/6 BMMCs and BALB/c BMMCs in resting and stimulated conditions with Pearson correlation coefficient $r = 0.86$ under the resting conditions and $r = 0.9$ under the stimulated conditions (Supplementary Fig. 4). The expression of the ID genes in resting BMMCs and the cytokine and chemokine genes in the resting and stimulated BMMCs was nearly identical (Supplementary Data 6). We chose the five-day time point based on our previous time-course experiments[10]. Five days after the initial culture with 4-OHT, the *Gata2*[−/−] MCs did not reduce their viability as measured by live cell counts[10] and cell death markers[11], while FcεR1α and c-KIT expression were reduced by on average 50%[10,11,43]. We

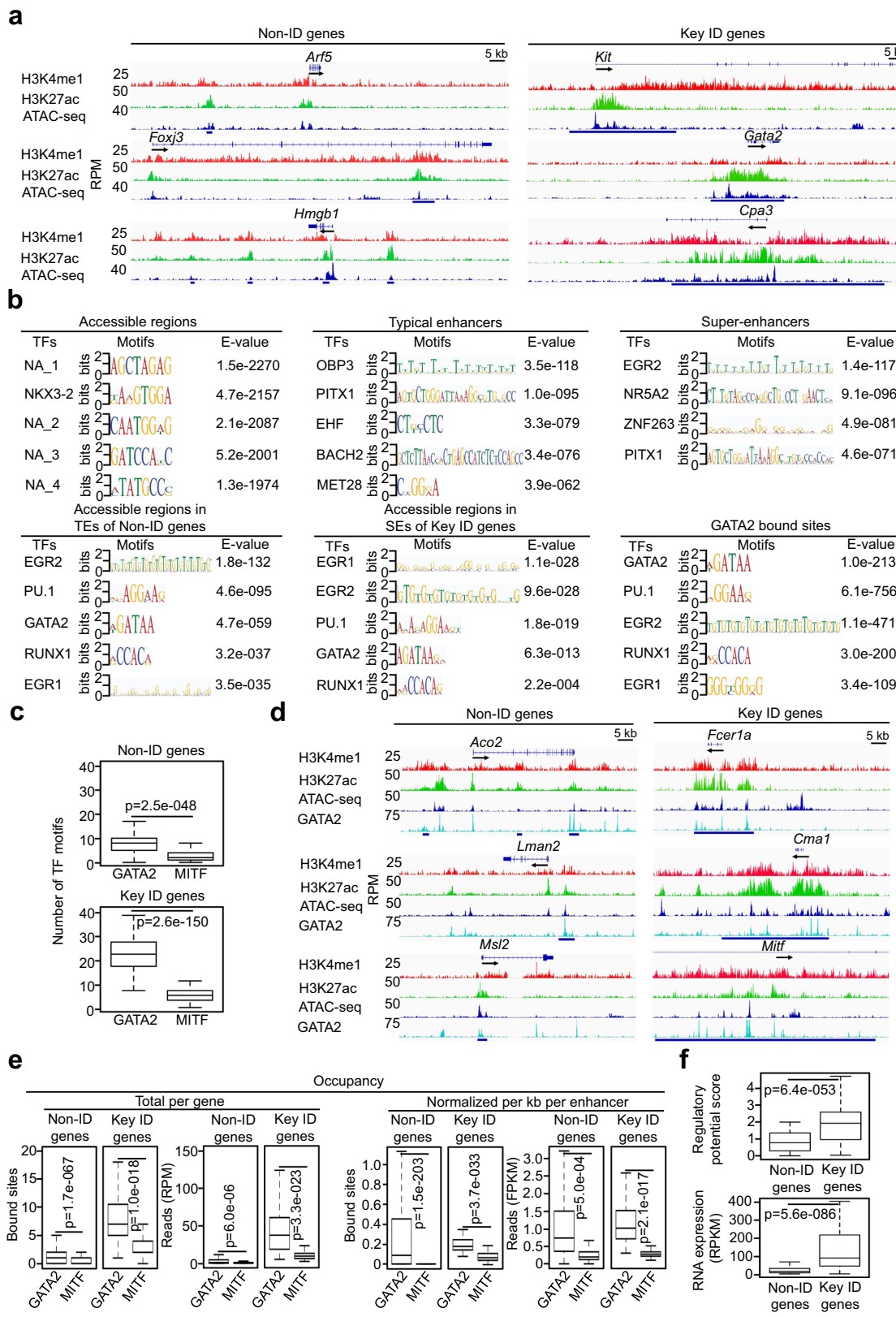

observed highly significant reductions (67%, $n = 2$ biological replicates) in chromatin accessibility at the super-enhancers compared with less (23%) reduction at the typical enhancers (Fig. 3a, Supplementary Data 7). H3K4me1 was reduced by 37% and 40% at the typical enhancers and super-enhancers ($n = 2$ biological replicates), respectively, while H3K27ac modification was reduced by

43% and 51% at the typical enhancers and the super-enhancers ($n = 2$ biological replicates), respectively (Fig. 3a). These data indicate that chromatin accessibility at the super-enhancers was more dependent on the presence of GATA2. Consistent with the reduction in chromatin accessibility at the super-enhancers, the key ID genes on average lost 44% mRNA expression, while the non-ID

**Fig. 2 The GATA2 binding motif and occupancy are enriched at the accessible regions within the super-enhancers of the key mast cell identity genes.**
**a** Representative tracks of the key ID genes and non-ID genes. **b** Enrichment analysis of TF binding motifs at the accessible regions, TEs and SEs separately or in the overlapped regions between the accessible regions and TEs or between the accessible regions and SEs. TF binding motif enrichment was also analyzed for the peaks from GATA2 ChIP-seq (GATA2-bound sites). NA means unknown motifs. **c** The numbers of GATA2 and MITF motifs per gene. **d** Representative tracks for the key ID and non-ID genes. **e** The GATA2 and MITF occupancy. Total per gene indicates the number of average bound sites and reads per gene. Data (**a**-**e**) represent two biological samples. **f** The regulatory potential scores calculated using GATA2-bound sites (upper panel) ($n = 3$ biologically independent samples). The RNA transcripts (lower panel). RPKM, reads per kilobase per million mapped reads. Data represent three biological samples. P-values were calculated by a two-tailed student's t test without adjustments. The IGV tracks are generated from one biological sample, representing two biological replicates with similar patterns. Middle line inside each box represents the median, upper and lower bounds of the box represent the third and first quartiles, respectively. Whiskers represent 1.5 times of the interquartile range.

genes did not lose significant mRNA expression (Fig. 3a, Supplementary Data 7).

We noticed that the requirement for GATA2 to maintain chromatin accessibility, H3K4me1 and H3K27ac modifications at enhancers varied from gene to gene. 100% of the super-enhancers of the key ID genes showed at least 46% reduction in chromatin accessibility, 9% reduction in H3K4me1 and 16% reduction in H3K27ac in the absence of GATA2. Notably, the super-enhancers of genes encoding MC proteases (*Tpsab1/Mcpt7*, *Tpsab2/Mcpt6*, *Tpsg1*) showed the greatest reductions in chromatin accessibility (80–99%), 60–79% reduction in H3K4me1 modification and 80–99% reductions in H3K27ac modification. Consistent with the marked reduction in chromatin accessibility, H3K4me1 and H3K27ac modifications, transcripts of these genes were reduced by 80–99% in the absence of GATA2 (Fig. 3b, lower panel). Lesser reductions in chromatin accessibility and histone modifications were observed at the super-enhancers of the genes that encode MC enzymes (*Hs3st1*, *Mmp27*, *Ddc*, *A4galt*, *Papss2*, *Ptpn2*, *Plau*, *Car8*, *Pde1c*). The super-enhancers at these genes exhibited 60–79% reduction in chromatin accessibility, 27–74% reduction in H3K4me1, 40–79% reduction in H3K27ac and 60–90% reduction in RNA transcripts (Fig. 3b, lower panel). We found a moderate correlation between reductions in chromatin accessibility and RNA transcripts (Pearson correlation coefficient, $r = 0.64$). In contrast, only six percent of the typical enhancers of the non-ID genes had reduced chromatin accessibility and histone modifications in the absence of GATA2 (Fig. 3b, upper panel). Representative tracks for the non-ID and key ID genes that exhibited the largest changes in the absence of GATA2 are shown in (Fig. 3c). Taken together, our data demonstrate that GATA2 binding promotes chromatin accessibility and histone modifications, with the strongest effects observed on chromatin accessibility and histone modifications at the super-enhancers of the key ID genes in resting MCs.

MCs are long-lived cells and expressed some of orthologous the self-renewal genes that have been defined in human embryonic stem cells and macrophages[44] (Supplementary Data 8). To explore whether GATA2 regulates the orthologous self-renewal genes, we analyzed GATA2 binding sites in these genes and found that binding of GATA2 to the enhancers of these genes (Supplementary Fig. 5). However, deletion of GATA2 did not affect the expression of these potential self-renewal genes (Supplementary Data 8).

**IgE receptor crosslinking induces expression of genes that encode signaling molecules, MC proteases, transcription factors, MC surface molecules, cytokines and chemokines.** Given that our data implicate GATA2 as important for transcribing genes that are constitutively expressed in resting MCs (and confer MC cell identity), we next investigated whether GATA2 regulates genes that are induced to perform specialized functions in

response to external stimuli. Although MCs respond to a host of stimuli including IL-18, IL-33 and complement 5a[45–47], IgE receptor crosslinking is the major driver of MC activation. We performed RNA-seq analysis on BMMCs ($n =$ three biological replicates) that were not stimulated or stimulated with IgE receptor crosslinking for two hours. We identified an average of 1089 genes ($n = 3$) that were expressed >2-fold in activated MCs relative to resting MCs (Fig. 4a, Supplementary Data 9). We verified the gene expression patterns of the selected top-ranking genes using qPCR (Supplementary Fig. 6).

Gene Ontology (GO) enrichment analysis of the IgE receptor crosslinking-upregulated (referred to hereafter as activation-induced) transcripts revealed that the upregulated genes were significantly enriched in gene sets that encode signaling molecules, transcription factors, proteases and other enzymes, cell surface molecules, and cytokines and chemokines (Fig. 4b, Supplementary Data 9). The most significantly enriched activation-induced genes were those encoding signaling molecules involved in the JAK-STAT signaling pathway, followed by the PI3K-Akt (*Sgk1* and *Cdkn1a*) and MAPK signaling pathways (*Mapkapk2*, *Map3k5* and *Map2k3*) (Fig. 4c, Supplementary Data 9). The next most enriched genes were cytokine and chemokine genes. We detected increases in *Il1a*, *Il4 Il5*, *Il6*, *Il13*, and *Tnf* mRNA in the activated MCs (Fig. 4c, Supplementary Data 9). Transcripts of chemokines *Ccl1*, *Ccl2*, *Ccl3*, *Ccl4* and *Ccl7* increased dramatically in activated MCs (Fig. 4c, Supplementary Data 9). Consistent with the transcript data, *Ccl1*, *Ccl2*, *Ccl3*, *Ccl4* and *Ccl7* protein levels were also dramatically increased (Supplementary Fig. 7). Another class of genes elevated in the activated MCs were TF genes. Among the upregulated TF genes, *Nfkb1*, *Nfatc1*, *Egr1*, *Egr2*, *Atf3* and *Ets1* have been reported to play critical roles in MC gene transcription in activated MCs. Notably, EGR1 and EGR2 are required for transcribing cytokine and chemokine genes in activated MCs[40,48]. These results suggest that upregulation of genes encoding signaling molecules, receptors and transcription factors constitute positive feedback loops that rapidly augment inflammatory responses by replenishing histamine and other inflammatory mediators, induce MC proliferation and promote MC survival.

We also observed that 280 genes were significantly suppressed following IgE receptor cross-linking (Supplementary Data 9). GO enrichment analysis revealed that the repressed genes were significantly enriched in gene sets involved in cell proliferation, signal transduction, apoptosis or enriched in genes encoding receptors (Supplementary Data 9).

**Super-enhancers are enriched in the key genes that respond to antigenic stimulation.** We analyzed enhancers associated with the activation-induced genes. Super-enhancers were enriched 4-fold in these genes compared to other genes expressed in MCs, representing a 21.2 % of all the activation-induced MC genes

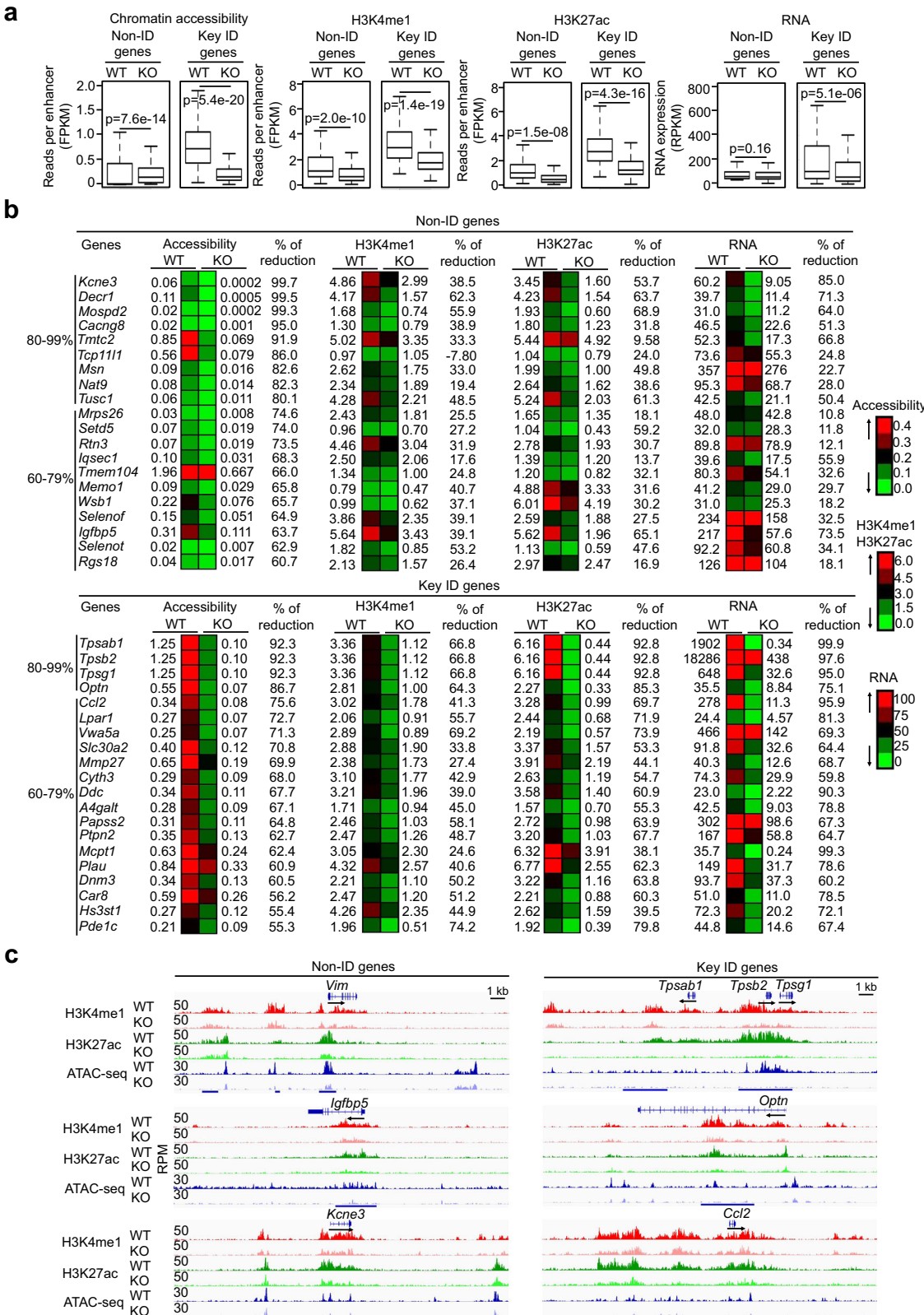

**Fig. 3 GATA2 promotes chromatin accessibility and H3K4me1 and H3K27ac modifications of histones associated with the MC genes. a** Calculated average reads at enhancers for the key ID genes and the non-ID genes in wild-type MCs (WT) and *Gata2⁻/⁻* MCs (KO). Middle line inside each box represents the median, upper and lower bounds of the box represent the third and first quartiles, respectively. Whiskers represent 1.5 times of the interquartile range. *P*-values were calculated by a two-tailed student's *t* test without adjustments. **b** Heatmap presentation of reads at the enhancers (FPKM) and RNA reads (RPKM). **c** Representative tracks. Data **a–c** represent two biological samples. The IGV tracks are generated from one biological sample, representing two biological replicates with similar patterns.

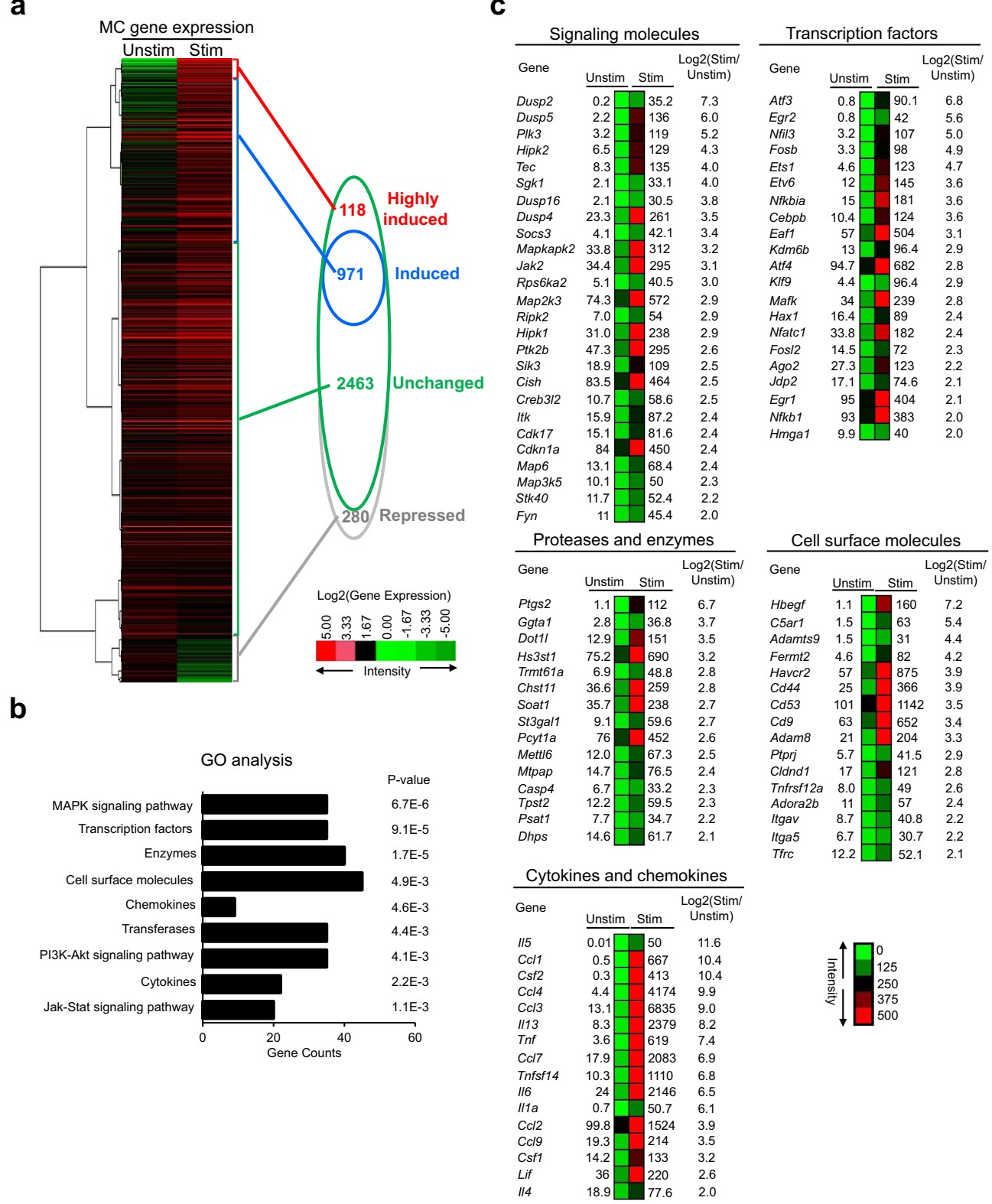

**Fig. 4 IgE receptor crosslinking induces expression of genes that encode signaling molecules, Proteases and enzymes, transcription factors, MC surface molecules, cytokines and chemokines. a** Heatmap presentation of the RNA transcripts after IgE receptor crosslinking for two hours. Unstim: unstimulated; Stim: stimulated. "Highly induced" means that the fold of induction (Stim vs Unstim) is higher than 10, "induced" means that the fold of induction ranges from 2 to 10, "unchanged" means that the fold of change is higher than 0.5 and lower than 2 and "repressed" means that the fold of change is lower than 0.5. **b** DAVID gene ontology enrichment analysis of the activation-induced genes. *P*-values were calculated by a one-side Fisher's exact test with the adjustment of Benjamini-Hochberg method. **c** Heatmap representations of top activation-induced genes. The numbers indicate RNA reads (RPKM). Data **a–c** represent three biological samples.

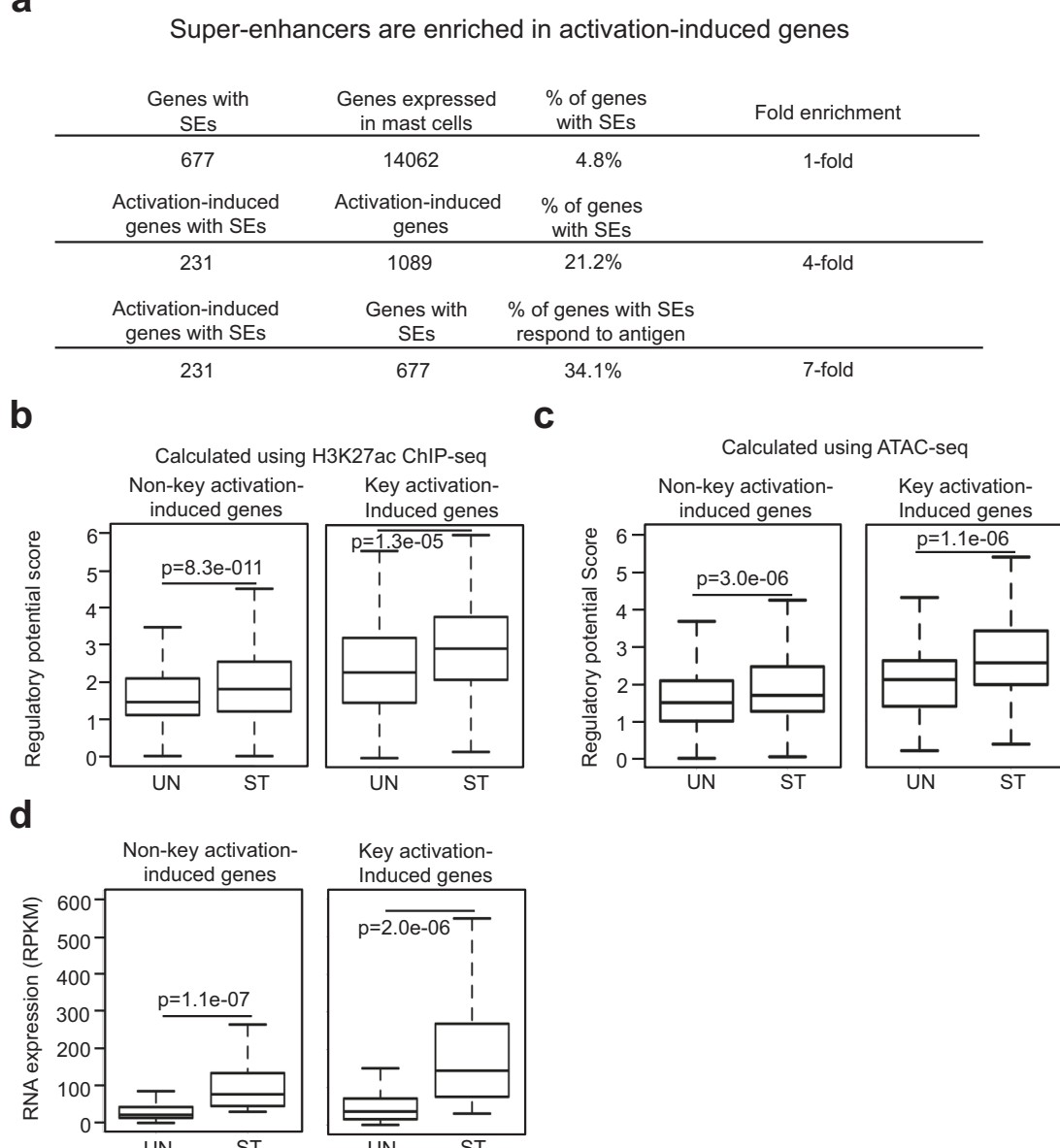

**Fig. 5 Super-enhancers are enriched in the key activation-induced genes. a** SEs are enriched in the key activation-induced genes. **b** The regulatory potential scores for TEs of the non-key activation-induced genes and SEs of the key activation-induced genes in resting (UN) or activated (ST) MCs were calculated either using H3K27ac ChIP-seq peaks (**b**) or using Omni-ATAC-seq peaks (**c**). **d** Average FPKM reads of RNA transcripts of the non-key activation-induced genes and key activation-induced genes in resting or activated mast cells (*n* = 3 biologically independent samples). Data **a**–**c** represent three biological samples. Middle line inside each box represents the median, upper and lower bounds of the box represent the third and first quartiles, respectively. Whiskers represent 1.5 times of the interquartile range. *P*-values were calculated by a two-tailed student's *t* test without adjustments.

(Fig. 5a, Supplementary Data 10). The activation-induced genes associated with super-enhancers represent a striking 34% of all MC genes associated with super-enhancers. We refer to the activation-induced genes with super-enhancers as the key activation-induced genes and the activation-induced genes with typical enhancers as the non-key activation-induced genes. The key activation-induced genes had 1.7-fold higher regulatory potential scores (calculated either using H3K27ac peaks or Omni-ATAC-seq peaks) than the non-key activation-induced genes in resting MCs (Fig. 5b, c). These genes showed a 1.2-fold increase in the regulatory scores when MCs were activated (Fig. 5b, c). The key activation-induced genes with higher regulatory scores were expressed 2-fold higher than the non-key activation-induced

genes in activated MCs (Fig. 5d). The finding that a high percentage of the activation-induced genes have super-enhancers suggests that a large portion of super-enhancers in MCs are programmed to respond to antigenic stimulation. Sequences within the super-enhancers likely include elements that facilitate responses to antigenic stimulation.

**Activation-induced genes exhibit "open" chromatin configurations before MCs encounter antigen and increase chromatin accessibility and H3K27ac modifications in response to IgE receptor crosslinking.** Cytokine and chemokine genes are expressed at low levels in resting MCs and upregulate dramatically in activated MCs. We examined the status of H3K4me1,

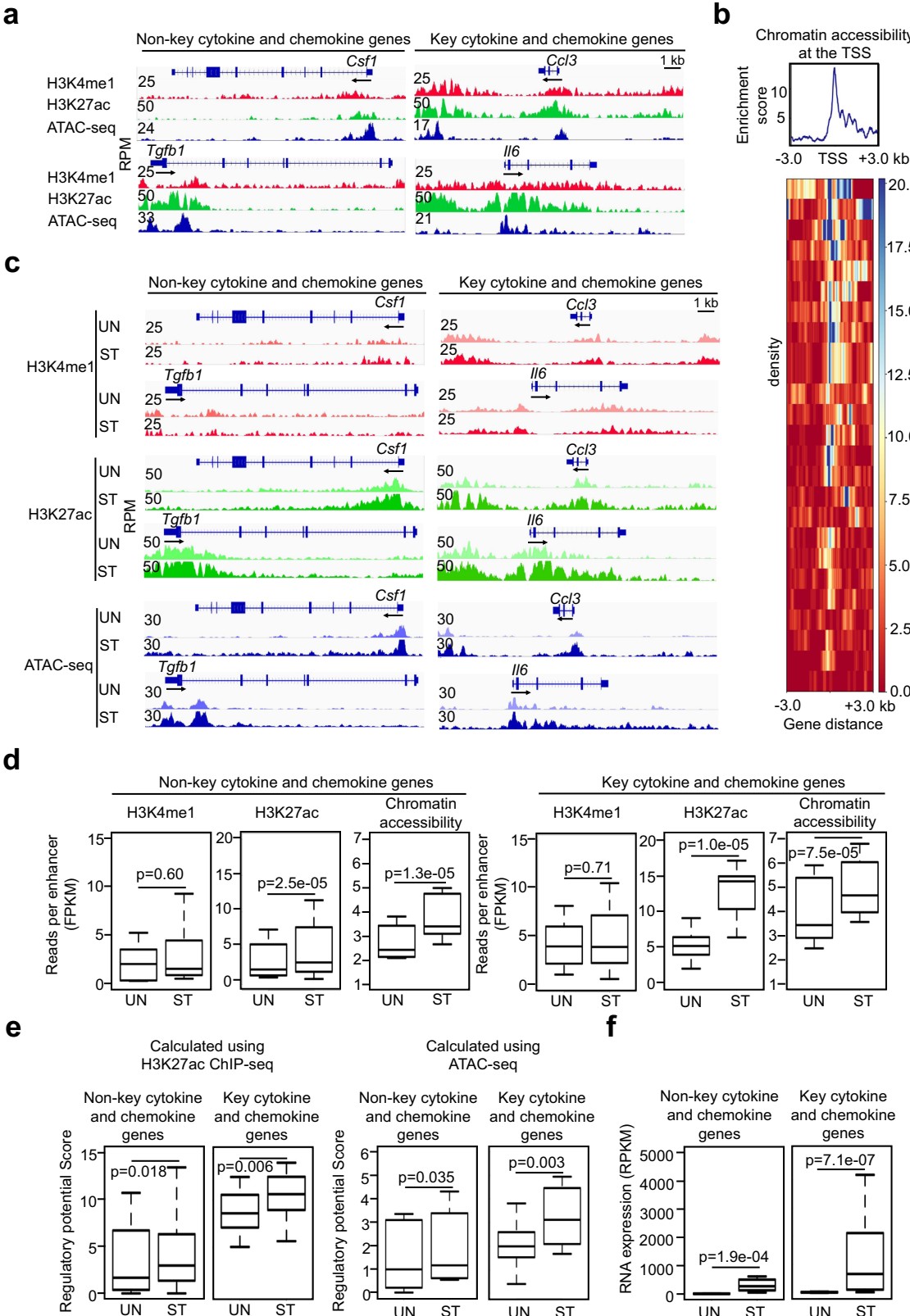

H3K27ac, and chromatin accessibility at the enhancers of activatable cytokine and chemokine genes in resting MCs. Representative tracks from one of the biological replicates shown in Fig. 6a provide evidence that the cytokine and chemokine genes with either typical or super-enhancers display enriched H3K4me1 modification before IgE receptor crosslinking. Surprisingly, these

genes also display enriched H3K27ac, a histone modification mark for actively transcribed genes, prior to stimulation by IgE receptor crosslinking (Fig. 6a). Moreover, these genes feature accessible chromatin at their proximal promoters, as measured by Omni-ATAC-seq (Fig. 6b). Our data support a model in which cytokine and chemokine genes are poised with an "open"

**Fig. 6 The activation-induced cytokine and chemokine genes show "open" chromatin configuration before MCs encounter antigen and show increased chromatin accessibility and H3K27ac modification in response to IgE receptor crosslinking. a** Representative tracks from resting MCs. **b** Enrichment distribution analysis of chromatin accessibility at the TSS of the cytokine and chemokine genes. **c** Representative tracks from resting (UN) and activated (ST) BMMCs. **d** Calculated average reads at enhancers per kb per enhancer in and activated BMMCs. **e** The regulatory potential scores were calculated using either H3K27ac ChIP-seq peaks or Omni-ATAC-seq peaks. Data **a–e** represent two biological samples. **f** Calculated average FPKM reads of RNA transcripts in and activated BMMCs ($n = 3$ biologically independent samples). The IGV tracks are generated from one biological sample, representing two biological replicates with similar patterns. Middle line inside each box represents the median, upper and lower bounds of the box represent the third and first quartiles, respectively. Whiskers represent 1.5 times of the interquartile range. $P$-values were calculated by a two-tailed student's $t$ test without adjustments.

chromatin configuration even before MCs encounter antigen. We refer to the cytokine and chemokine genes with super-enhancers as the key cytokine and chemokine genes.

Two hours after IgE receptor crosslinking, we observed that H3K27ac, but not H3K4me1, was significantly increased at the typical enhancer and super-enhancer regions (Fig. 6c). Concurrently, chromatin accessibility at both types of enhancers was also significantly increased (Fig. 6c). Numbers of normalized H3K27ac reads and Omni-ATAC-seq reads increased 3.0-fold and 1.7-fold (calculated from two biological samples), respectively, at the super-enhancers of the key cytokine and chemokine and 1.5-fold at typical enhancers of the non-key cytokine and chemokine genes after IgE receptor crosslinking (Fig. 6d, Supplementary Data 11). The regulatory potential scores also increased after stimulation, as evidenced by increased H3K27ac or Omni-ATAC-seq peaks (Fig. 6e left, right panel, respectively). The key cytokine and chemokine genes with higher regulatory potential scores were expressed in 8.5-fold higher amounts of mRNA relative to the non-key cytokine and chemokine genes (calculated from three biological samples) (Fig. 6f). Enhancers that showed increased chromatin accessibility and H3K27ac modification in response to IgE receptor crosslinking are denoted as activation-induced enhancers.

As shown in Fig. 4, there were 1089 genes that responded to IgE receptor crosslinking with increased mRNA expression. We found that these genes also had an "open" chromatin configuration before MCs encountered antigenic stimulation. Upon IgE receptor crosslinking, H3K27ac modification and chromatin accessibility, but not H3K4me1, were significantly increased at typical enhancers and super-enhancers. Increased H3K27ac also correlated positively with increased regulatory potential scores and increased mRNA expression (Supplementary Fig. 8).

**GATA2 primes the activation-inducible enhancers to respond to antigenic stimulation.** To determine whether GATA2 plays a role in regulating the activation-induced enhancers, we conducted TF binding motif analysis on regions marked by activation-increased H3K27ac modification to identify motifs that promote gene transcription. The accessible regions that overlapped with the activation-induced enhancers are highlighted in Fig. 7a. The activation-induced accessible regions located outside of the regions with H3K27ac modification regions could contain TF binding motifs that repress gene transcription and are included in Supplementary Data 5. TF binding motifs enriched in these regions included HIC2, DUX4, KLF4, E2F8 and HHEX (Supplementary Fig. 3, right panel), the majority of which have been reported to show transcriptional repressive activities[49–53]. EGR2, AP1 and GATA2 binding motifs were significantly enriched within the overlapped accessible regions (Fig. 7a). Enrichment of these TF binding motifs was not found when the activation-induced accessible regions, the activation-induced typical or the activation-induced super-enhancers were analyzed without intersecting the accessible regions within the activation-induced enhancers (Supplementary Fig. 9).

To distinguish the activation-induced enhancers in resting MCs from those in activated MCs, we refer to the former as the activation-inducible enhancers. To investigate a potential requirement for GATA2 in regulating the activation-inducible enhancers, we counted the number of GATA2 and MITF binding motifs within these enhancers. The number of GATA2 binding motifs was significantly higher than that of MITF binding motifs at both types of activation-inducible enhancers (Fig. 7b). To verify that GATA2 binds directly to the activation-inducible enhancers, we performed GATA2 ChIP-seq on resting BMMCs ($n = 2$) and used the downloaded MITF ChIP-seq data generated from resting BMMCs[38]. Because recent studies have shown that chromatin accessibility at super-enhancers can change in just a few hours after stimulation, we also performed GATA2 ChIP-seq on BMMCs that were stimulated with IgE receptor crosslinking for two hours ($n = 2$). We found that the total frequencies and densities of GATA2 binding sites were higher at the inducible super-enhancers than at the inducible typical enhancers (4.2- and 5.1-fold higher, respectively, Fig. 7b, middle panel). We found the normalized densities and reads of GATA2-bound sites per enhancer were also higher at the activation-inducible super-enhancers (1.5- and 2.1-fold higher, respectively, ($n = 2$), Fig. 7b, right panel). We did not observe significant changes in GATA2 binding both at the typical or the super-enhancers before or after stimulation. These data support that GATA2 binds to both typical and super-enhancers in resting MCs before they are stimulated and that GATA2 occupancy at the enhancers of are not significantly altered by antigenic stimulation.

To determine whether the formation of activation-induced enhancers is critically dependent on GATA2 binding, we deleted the *Gata2* gene in the inducible *Gata2* knockout BMMCs ($n = 2$). Figure 7c shows that five days after the initial treatment with 4-OHT, 93% of $Gata2^{-/-}$ MCs expressed the IgE receptor FcεR1 at 56% lower levels than WT MCs as measured by MFI (Fig. 7c, Supplementary Fig. 10). Gene transcripts for the *Fcer1a* (encodes IgE receptor α chain), *Ms4a2* (encodes IgE receptor β chain), and *Fcer1g* (encodes IgE receptor γ chain) genes were reduced by 38%, 52% and 45%, respectively, in resting $Gata2^{-/-}$ MCs (Supplementary Data 12). Gene transcripts for some signaling molecules involved in IgE signaling were reduced by 4–66%, while gene transcripts for other signaling molecules involved in IgE signaling increased 1.1 to 6.7-fold (Supplementary Data 12). The resting $Gata2^{-/-}$ MCs showed slight reductions in chromatin accessibility and H3K27ac modification at the activation-inducible enhancers (Fig. 7d, Supplementary Data 13). Remarkably, $Gata2^{-/-}$ MCs with only partially impaired IgE receptor signaling completely failed to increase chromatin accessibility and H3K27ac modification at activation-inducible enhancers in $Gata2^{-/-}$ MCs (Fig. 7d), indicating that GATA2 is required for the formation of the activation-induced enhancers. Not only did the activation-inducible enhancers fail to be activated in MCs lacking GATA2, but these enhancers exhibited diminished chromatin accessibility in response to IgE receptor crosslinking (Fig. 7d, left panel, $P < 0.01$, $n = 2$). These data indicate that signals triggered by the

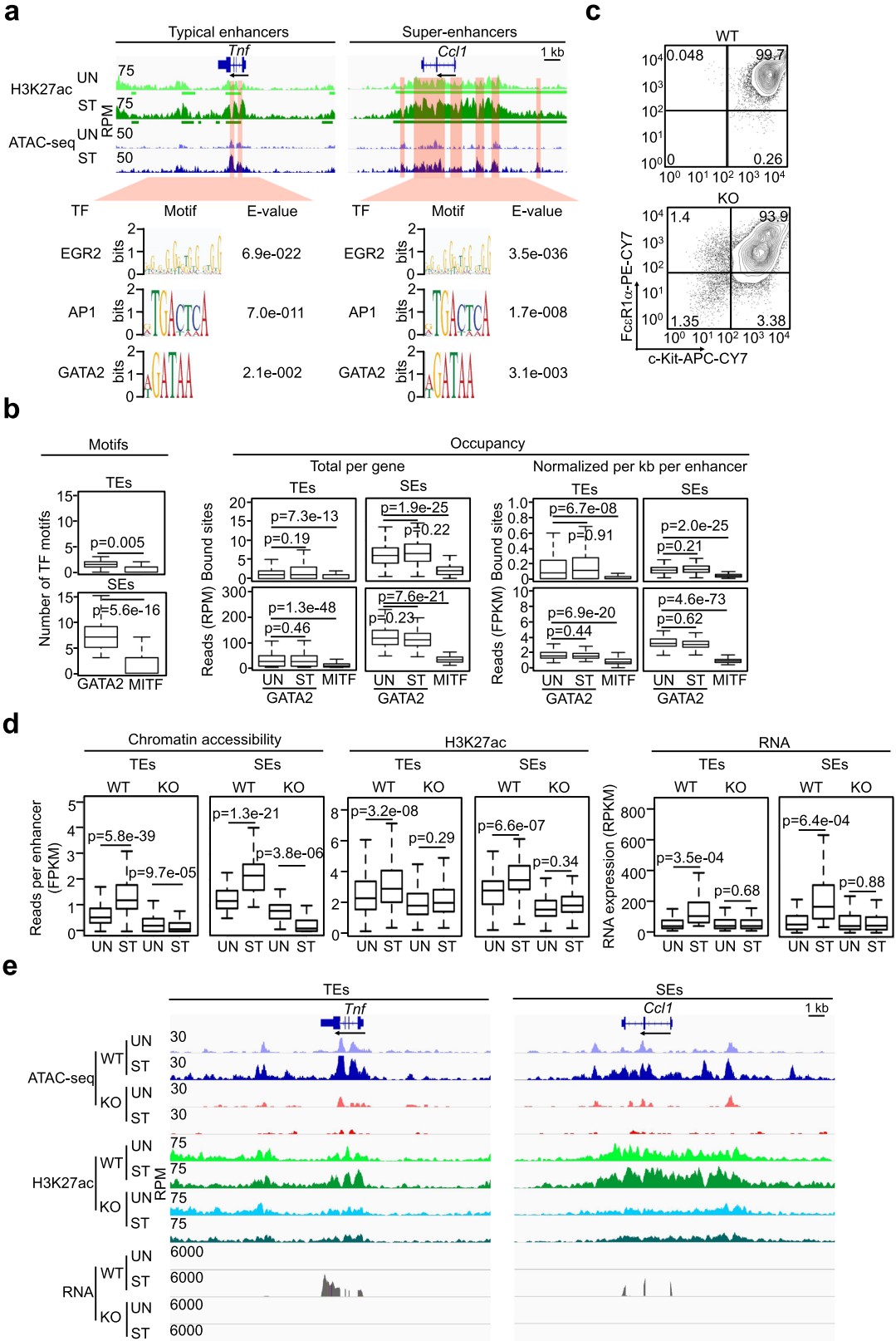

stimulation were delivered to the activation-inducible super-enhancers by a mechanism involving GATA2. Moreover, *Gata2*⁻/⁻ MCs completely failed to upregulate gene transcription in response to IgE receptor crosslinking (Fig. 7d, Supplementary Data 13). Representative tracks from one of the biological replicates for reductions in chromatin accessibility, histone modifications and

RNA transcripts in *Gata2*⁻/⁻ MCs are shown in (Fig. 7e). To determine whether *Gata2*⁻/⁻ MCs were functionally responsive, we examined the responsiveness of *Gata2*⁻/⁻ MCs to IgE receptor crosslinking. *Gata2*⁻/⁻ MCs released similar amounts of β-hexosaminidase under resting conditions and released 74% less β-hexosaminidase after IgE receptor crosslinking

**Fig. 7 GATA2 primes the activation-inducible enhancers to respond to antigenic stimulation. a** TF binding motifs enriched in the overlapped regions between the regions with the activation-induced H3K27ac modification and the regions with the activation-induced accessibility (highlighted). H3K27ac ChIP-seq and Omni-ATAC-seq were prepared from resting (UN) and activated mast cells (ST). **b** The number of GATA2 and MITF binding motifs (left panel) and occupancy (total bound sites and reads per gene, middle panel) and normalized bound sites and reads per kb per enhancer (right panels) ($n = 2$ biologically independent samples). **c** Flow cytometry analysis of the WT and *Gata2* KO BMMCs at 5 days after 4-OHT treatment. **d** Calculated average reads at enhancers and RNA transcripts for WT and *Gata2* KO MCs under resting or activated conditions. Two biological samples for each NGS sequencing under each condition. **e** Representative tracks for cells and treatments indicated in **d**. The IGV tracks are generated from one biological sample, representing two biological replicates with similar patterns. Middle line inside each box represents the median, upper and lower bounds of the box represent the third and first quartiles, respectively. Whiskers represent 1.5 times of the interquartile range. *P*-values were calculated by a two-tailed student's *t* test without adjustments.

(Supplementary Fig. 11a). *Gata2$^{-/-}$* MCs completely failed to produce IL-6 protein and only produced 7% TNFα protein compared to WT BMMCs in response to IgE receptor crosslinking (Supplementary Fig. 11b). Our results provide evidence that GATA2 binding to the activation-inducible enhancers of MCs prior to IgE receptor crosslinking is required for priming the activation-inducible enhancers to respond to antigenic stimulation.

## Discussion

Although we and others have demonstrated that GATA2 is essential for the differentiation of MC progenitor cells into the MC lineage[3,4] and the maintenance of the identity of fully committed MCs, it has not been investigated how GATA2 regulates enhancers of its target genes in MCs. We identified typical and super-enhancers of MC genes in resting and antigen-stimulated MCs. We found that super-enhancers are enriched with GATA2 binding motifs and bound sites, supporting the role of GATA2 as a critical regulator of chromatin accessibility and histone modifications associated with active enhancers. These conclusions are consistent with our previous observation that GATA2 is expressed in relatively large amounts in MC progenitors[10]. GATA2 is a member of a special class of proteins that accesses its DNA target sites in the context of nucleosomes, whereas other factors require more accessible DNA for binding[54]. Once bound, these factors collaborate to promote chromatin accessibility and gene transcription. Indeed, our observation that GATA2 is bound to the typical and super-enhancers in resting BMMCs suggests its roles as an early initiator of gene transcriptional patterning in MC development.

Super-enhancers are often associated with genes that confer cell identities[17,18]. However, mechanisms by which super-enhancers regulate cell identity genes have not been completely delineated. Furthermore, super-enhancers might function differently in different cell types, dependent on which transcription factors are being expressed in the particular cell types. Super-enhancers that act in MC differentiation and MC responsiveness to antigenic stimulation have not been investigated. Our findings reveal a mechanism of how GATA2 contributes to the high levels of target gene transcription associated with MC identity and robust responsiveness to antigenic stimulation. We demonstrated that the same set of binding motifs of GATA2 and co-TFs PU.1, RUNX1, EGR1 and EGR2 are enriched at the accessible regions of both the typical enhancers and super-enhancers. However, the number of GATA2 binding motifs and occupied sites, but not those of the other MC LDTF, MITF, are observed at several fold higher frequencies at super-enhancers relative to typical enhancers. In fact, the super-enhancers in the key ID genes not only have more GATA2-bound sites per gene overall, but also have higher densities of GATA2-bound sites relative to typical enhancers. Moreover, super-enhancers have higher regulatory potential scores. Key ID genes with higher regulatory potential scores expressed much higher levels of mRNA. These results

support a model in which GATA2 promotes higher levels of target gene transcription by binding to a larger number of clustered binding sites available at the super-enhancers of key ID genes. This model is consistent with modes of action of other LDTFs in other cell types. For example, in ES cells, the relative occupancy of binding sites for the TFs KLF4 and ESRRB, but not OCT4, SOX2, or Nanog bound sites, increase at the super-enhancers of the key ES identity genes compared to the typical enhancers[17]. In B cells, E2A binding densities[17], but not PU.1-, FOXO1- or EBF1-bound sites, are present in increased numbers at the super-enhancers of the key B cell identity genes[17].

We identified a class of enhancers that mediate responses to antigenic stimulation. These activation-induced typical and super-enhancers have increased chromatin accessibility and H3K27ac modification but do not have increased H3K4me1 modification in response to antigenic stimulation. We propose that binding of GATA2 to the activation-inducible enhancers in resting MCs primes these regions by recruiting the SDTFs that are activated by signals triggered by antigen. Our results provide evidence that the assembly of these activation-induced enhancers is GATA2-dependent. In the absence of the *Gata2* gene, activation-inducible enhancers failed to be triggered after antigenic stimulation. Our RNA-seq data obtained from the inducible *Gata2* gene deletion experiments further support this GATA2 priming model. We found that in the absence of the *Gata2* gene, IgE receptor crosslinking completely failed to upregulate gene transcription of those genes that normally would be upregulated. The failure to upregulate gene transcription in response to antigenic stimulation cannot be explained solely by the partial reduction in IgE receptor signaling capacity. In fact, further diminution in chromatin accessibility at the activation-inducible super-enhancers after IgE receptor crosslinking argues that the partially reduced IgE receptors on *Gata2$^{-/-}$* MCs were capable of delivering signals. Moreover, our published data showed that phorbol myristate acetate (PMA) and ionomycin stimulation, which bypass the need for FcεR1α to activate MCs, also failed to upregulate the genes that normally would respond to antigenic stimulation[10], suggesting that without GATA2 priming, the activation-inducible enhancers fail to be triggered and, as a result, they fail to recruit SDTFs. The failure of triggering the activation-inducible enhancers might lead to the inability of MCs to upregulate gene transcription in response to antigenic stimulation.

TF motif enrichment analysis of the activation-induced enhancers supports roles of the SDTFs AP1 and EGR2, which bind the activation-inducible enhancers in response to antigenic stimulation to recruit the co-activator P300. In turn, P300 adds additional H3K27ac modifications. This proposed model is consistent with recent findings from a study on human MCs. Cildir and colleagues reported that IgE receptor crosslinking induces Ca$^{2+}$-dependent chromatin domains in a genome-wide fashion[55]. AP1 binding motifs are enriched at the Ca$^{2+}$-dependent chromatin domains. Wu et al. showed that EGR2, which is rapidly increased after IgE receptor crosslinking, directly binds to the *Ccl1* chemokine gene

promoter and is required for CCL1 production but not for CCL3 and CCL9 production[40]. Our data confirmed that *Egr2* mRNA rapidly increased after IgE receptor crosslinking on BMMCs, supporting a role for EGR2 in regulating genes that respond to antigenic stimulation. Because there are a large number of zinc finger family TFs in MCs that potentially bind consensus EGR2 binding motifs, the identity (ies) of the exact SDTFs that bind to the induced regions after IgE receptor crosslinking need to be assessed on an individual basis. Heinz and colleagues reported that strain-specific PU.1, CEBPα and AP1 binding sites co-localize with regions marked with H3K4me2 and H3K27ac modifications. They showed that PU.1, CEBPα and AP1 prime enhancers to respond to macrophage activation signals in a strain-specific manner. The primed enhancers require additional binding of SDTF P65, a component of NFκB, to be fully functional[56]. Their work illustrates a mechanism by which LDTFs prime enhancers to respond to external stimulations. However, the work was conducted prior to the advancement of next generation sequencing and only a few genes in one cell type were observed. Our proposed model is consistent with these studies of macrophages, but also illuminate how MCs prepare their enhancers to respond to antigenic stimulation. Together, our work adds evidence supporting the priming of enhancers by LDTFs in multiple cell types.

Our studies raise the question of whether the activation-induced H3K27ac modification is used as an epigenetic mark to record the history of activation-induced gene transcription. Cytokine and chemokine gene transcription returned to background levels 24 h after antigenic stimulation. The rapid disappearance of cytokine and chemokine mRNA posts a challenge for accurate measurements of cytokine and chemokine gene expression in patients experiencing acute inflammation. A 40% reduction in the activation-induced H3K27ac modification 5 days after the *Gata2* gene was deleted allows us to estimate that activation-induced H3K27ac potentially persists for 10 days after the SDTFs AP1 and EGR2 leave the activation-induced enhancers. However, activation-induced epigenetic marks could persist much longer under chronic disease conditions. Thus, the delayed removal of H3K27ac at the activation-induced enhancers may serve as a useful epigenetic marker of previous cytokine and chemokine gene transcription following an episode of acute inflammation and during chronic inflammation.

IgE receptor crosslinking induces profound changes in gene expression in MCs. While a small percentage of upregulated genes have been reported before, the majority of upregulated genes have not been reported previously. The activation-induced gene expression enables MCs to perform more robust functions and recruit other types of cells to sites of inflammation. Notably, the increased expression of the *Il1r1* gene encoding IL-33 receptor promotes IL-33 signaling, leading to further upregulation of cytokine genes and chemokine genes, including *Il13*[57]. Increased expression of the signaling proteins LYN, JAK1, PIK3CD, and MAPKAPK3 strengthen potential signaling through major signaling pathways, including MAKP, PI3K, and JAK-STAT. Increased expression in the *Bcl2* gene suggests that the activated MCs enhance MC survival. Increased expression in the *Nlrp3* (Cryopyrin) gene suggests that the MCs can have a stronger ability to release pro-inflammatory cytokines such as IL-1β and IL-18 in response to bacterial infection and other insults. We also observed that a number of genes enriched in gene sets involved in cell proliferation, signal transduction, apoptosis or enriched in genes encoding receptors were significantly suppressed following IgE receptor cross-linking. Downregulation of these genes allow activated MCs to return to resting state. Notably, downregulation of the *Kit* (encode the receptor for SCF), *Gata2*, *Stat5b* by IgE crosslinking suggest that activated MCs become less sensitive to SCF stimulation. Downregulation of the *Ma4a2* (encode IgE receptor β chain),

*Pip5k1c*, *Pik3cg*, *Pi3kcd* genes could lead to termination of signals triggered by IgE receptor crosslinking. Downregulation of the apoptotic genes promote MC survival after activation. The mechanisms by which IgE crosslinking downregulate these genes are not clear. Around 8.1% of the activation-induced accessible regions in response to IgE receptor cross-linking was located outside the enhancer regions with H3K27ac modification. We found that these activation-induced accessible regions are enriched in transcription repressor binding motifs. A significant number of these repressive TF binding motifs target genes were repressed by IgE receptor crosslinking. Further studies using our combined strategy of TF binding motif enrichment should identify transcription repressors induced by antigenic stimulations.

In summary, MCs are the cause of MC disorders. MCs remodel their chromatin landscapes in response to antigen exposure and do not die after antigenic stimulation and degranulation[58,59], posing great challenges for effectively treating MC-mediated diseases. The detailed molecular analysis presented here provides information that may aid in the development of effective preventions and therapeutic strategies.

## Methods

**Animals**. Balb/cJ mice (000651) were purchased from the Jackson Laboratory (Bar Harbor, ME). The inducible *Gata2* conditional knockout mice (*Gata2^{f/f}Rosa^{Yfp/Yfp}TgCreErt2^{hemi}*) or wild-type control mice (*Gata2^{+/+}Rosa^{Yfp/Yfp}TgCreErt2^{hemi}*) on 129 genetic background backcrossed to C57BL/6 genetic background for 5 generations. These mice were generated by crossing *Gata2^{f/f}* mice to Cre activity reporter RosaYfp/Yfp mice and inducible Cre mice[10]. Six to twelve-week old male or female mice were used for experiments. Mice were kept under the standard housing conditions: 72 to 74 °F ambient temperature, 50-60% humidity and 12 h dark/light cycle. All animal experiments were conducted according to protocols approved by the National Jewish Health Institutional Animal Care and Use Committee (Protocol number: AS2703-02-23).

**Cell culture**. Bone marrow-derived mast cells (BMMCs) were cultured from bone marrow cells of Balb/c mice in Iscove's DMEM (10016CV, Corning™ cellgro™, Manassas, VA) plus 10% FBS,100 units/mL penicillin, 100 μg/mL streptomycin, and 2 mM beta-mercaptoethanol in the presence of 20 ng/mL IL-3 for four weeks. Over 99% of BMMCs were mast cells as determined by FACS analysis (FcεRIα+ c-Kit+).

**Inducible deletion of *Gata2* gene in BMMCs**. For experiments that examined the role of GATA2, BMMCs were cultured from bone marrow cells of inducible *Gata2* conditional knockout (*Gata2^{f/f}Rosa^{Yfp/Yfp}TgCreErt2^{hemi}*) mice or wild-type control (*Gata2^{+/+}Rosa^{Yfp/Yfp}TgCreErt2^{hemi}*) mice on 129 genetic background backcrossed to C57BL/6 genetic background for 5 generations. BMMCs were treated with 25 nM 4-hydroxytamoxifen (4-OHT; H7904-5MG, Sigma-Aldrich, St. Louis, MO) for five days[10,43]. Expression of FcεRIα, and c-Kit on mast cells were analyzed by Flow cytometry using 1:200 diluted fluorochrome labeled antibodies (PE-CY7–conjugated anti-FcεRIα, MAR-1 Biolegend #134317; allophycocyanin-CY7–conjugated anti–c-Kit, 2B8, Biolegend, #105825). The YFP+ cells were FACS-sorted and deletion of the *Gata2* gene was determined by PCR to be greater than 99%.

**BMMC stimulation**. BMMCs were sensitized with 1 μg/mL IgE anti-2, 4, 6-Trinitrophenyl (TNP) antibody (IGEL 2a, ATCC, Manassas, VA). Twenty-four hours later, the cells were challenged with 100 ng/mL TNP-BSA (T-5050-100, LGC Biosearch Technologies, Novato, CA) for 2 additional hours before the cells were collected for further analysis.

**Chromatin Immunoprecipitation and ChIP-seq**. $1 \times 10^7$ BMMCs that were not treated or treated with IgE receptor crosslinking were fixed with 1% formaldehyde (PI28908, Thermo Fisher Scientific), sonicated by using the Covaris S220 Focused-ultrasonicator in the SDS lysis buffer (1% SDS, 10 mM EDTA, 50 mM Tris.HCl pH8) and pre-cleared with Protein A Beads at 4 °C for 1 h according to established protocols[43]. The samples were incubated with 10 μg of following antibodies (1:100 dilution): anti- H3K4me1 antibody (ab8895, Abcam, Cambridge, MA), anti-H3K27ac antibody (ab4729, Abcam) or anti-GATA2 antibody (ab22849, Abcam) at 4 °C overnight and then with protein A agarose/salmon sperm DNA slurry (Millipore, Cat# 16-157) at 4 °C for 1 h. The beads were washed and eluted as described[43]. The crosslinking of eluted immunocomplexes was reversed and the recovered DNA was recovered using a QIAGEN QIAquick PCR purification kit (Qiagen, Valencia, CA). ChIP-seq library was prepared using TruSeq ChIP Library Preparation Kit (IP-202-1024, Illumina, San Diego, CA) according to the manufacturer's instructions. Briefly, 10 ng of ChIPed DNA was converted into blunt-

ended fragments. A single adenosine nucleotide was added to the 3' ends of the blunt-ended fragments before ligation of indexing adapters to the adenylated 3' ends. The ligated products were purified, size-selected and PCR amplified according to the manufacturer's instructions. The quality and quantity of the DNA library were assessed on an Agilent Technologies 2100 Bioanalyzer. Paired-ended sequencing was performed on an Illumina NovaSEQ6000 platform.

**RNA-seq**. To prepare RNA samples for RNA-seq, total RNA was isolated from $1×10^6$ BMMCs that were untreated or treated with IgE receptor crosslinking using the RNeasy mini kit (Qiagen). RNA was treated with DNase I according to manufacturer's instructions. The quality of RNA was analyzed using the Agilent Technologies 2100 Bioanalyzer. RNA-seq libraries were prepared using the NEB-Next Ultra II RNA Library Prep Kit for Illumina (E7770S, New England Biolabs, Ipswich, MA). Briefly, polyA mRNA was purified using the NEBNext Poly(A) mRNA Magnetic Isolation Module (NEB, E7490). The isolated mRNA was fragmented, reverse transcribed into cDNA, end-repaired, and adapter-ligated. The ligated samples were USER enzyme digested and PCR amplified and purified using AMPure-XP beads (Beckman Coulter Life Sciences, A63881, Indianapolis, IN). The quality and quantity of RNA libraries were assessed on an Agilent Technologies 2100 Bioanalyzer. Paired-ended sequencing of the RNA libraries was performed on an Illumina NovaSEQ6000 platform. The RNA-seq related to *Gata2* deletion experiments was performed by Novogene Corporation Inc., Sacramento, CA.

**Omni-ATAC-seq**. Omni-ATAC-seq was performed according to the established method[32]. Briefly, 50,000 BMMCs that were either untreated or treated with IgE receptor crosslinking were spun down and washed once with cold PBS. The cells were resuspended in 50 μl cold ATAC-RSB-lysis buffer (10 mM Tris-HCl pH 7.4, 10 mM NaCl, 3 mM MgCl$_2$, 0.1% NP-40, 0.1% Tween-20 and 0.01% Digitonin) and incubated for 3 minutes. The lysis buffer was immediately washed out with 1 mL ATAC-RSB buffer (10 mM Tris-HCl pH 7.4, 10 mM NaCl, 3 mM MgCl$_2$ and 0.1% Tween-20). The cell pellet was resuspended in 50 μl transposition mix (25 μl 2X TD buffer, 2.5 μl transposase (Illumina, FC-121-1030), 16.5 μl PBS, 0.5 μl 1% digitonin, 0.5 μl 10% Tween-20, 5 μl H$_2$O) and incubated at 37 °C for 30 minutes. The reaction was stopped by adding 2.5 μl of 0.5 M EDTA pH 8 and transposed DNA was purified using Qiagen MiniElute PCR purification kit (Qiagen). Purified DNA was amplified using the following condition: 72°C for 5 min, 98 °C for 30 s, and 7 cycles: 98 °C for 10 s, 63 °C for 30 s, 72 °C for 1 min. The amplified libraries were purified, size-selected, and the quality and quantity of libraries were assessed on an Agilent Technologies 2100 Bioanalyzer. The pair-ended sequencing of DNA libraries was performed on an Illumina NovaSEQ6000 platform.

**Reproducibility metrics**. Reproducibility of the ChIP-seq and Omni-ATAC-seq data was determined by using the deepTools 3.3.0[60] multiBamSummary and plotCorrelation functions. Briefly, the aligned read coverages were computed for consecutive bins of 10 kb size across the mm10 genome from BAM files generated from two biological samples by using multiBamSummary. The similarity in the mapped reads in the bins between two biological samples were analyzed by plot-Correlation. R values were calculated by using Pearson correlation coefficient. The reproducibility of the RNA-seq data was calculated by using the ggscatter function in R package ggpubr (version 0.2). The similarity of reads mapped to genes from two biological samples (FPKM, generated by using Cufflink) was estimated using Pearson correlation coefficient analysis. qPCR and ELISA data represent the results pooled from three independent experiments.

**Assigning enhancers to genes**. Genomic Regions Enrichment of Annotations Tool (GREAT, version 4.0.4) was used to assign the regulatory regions identified by H3K27ac ChIP-seq and Omni-ATAC-seq to their putative target genes[61] with the following setting: Species Assembly, Mouse: GRCm38; Basal plus extension, proximal 5.0 kb upstream and 1.0 kb downstream, plus distal up to 100 kb. The expression of putative target genes was analyzed further by comparison with our RNA-seq expression dat. Only the genes with RPKM higher than 30 were considered as enhancer associated target genes.

**Analysis of super-enhancers and typical enhancers**. The adapter sequences in raw reads (average 24.0 million reads, two biological replicates for each group) were trimmed using Trimmomatic (version 0.33)[62] and the quality of resulting sequence data was analyzed by FastQC (version 0.11.9). The trimmed reads were aligned to the mm10 reference genome (GRCm38) using Bowtie2 (version 2.3.5.1)[63]. Peak calling was performed using MACS2 (version 2.1.2) with default parameters with the q-value cut-off of 0.05[64]. Reads were presented as the read counts per million uniquely mapped reads. The analyzed sequence data was visualized using Integrative Genomics Viewer (IGV). H3K27ac ChIP-seq data were used for identifying typical and super-enhancers using ROSE software[17]. The H3K27ac ChIP-seq peaks in resting and activated mast cells were stitched within 12.5 kb of each other and excluding 2.5 kb upstream and downstream of the known transcription start sites (TSS). The combined H3K27ac reads within stitched regions were plotted in a ranked enhancer order. Super-enhancers were defined as the enhancers above the inflection point and the rest were defined as typical enhancers. Genomic distributions of the typical enhancers and super-enhancers were analyzed by the R package ChIPseeker (version 1.18.0)[65].

Average total numbers of typical enhancers and super-enhancers were calculated by dividing total numbers of typical enhancers and super-enhancers found in two replicates with two. Representative ChIP-seq IGV tracks shows in the figures were generated from biological sample 1 and are similar to those in biological sample 2. The GATA2 ChIP-seq peaks generated from resting BMMCs (Fig. 2) were downloaded from Gene Expression Omnibus (GSE48086). The GATA2 ChIP-seq data used in Fig. 7 was generated in our laboratory (GSE145544).

**Calculation of reads and bound sites at enhancers**. Enhancers were identified using ROSE (version 1). The H3K27ac ChIP-seq peaks were assigned to their putative target genes using GREAT. The enhancer reads were counted by using Bedtools multicov (version 2.29.2)[66]. Reads at enhancers per gene were calculated as the sum of reads at all enhancers assigned to a gene in two biological replicates and were represented as average reads per million nucleotides mapped (RPM). Normalized reads per enhancer were calculated by dividing the reads at enhancers per gene with a normalized size of enhancers, which is equal to the average sizes of enhancers assigned to a gene divided by 1000 bp. The data were represented as fragments per kilobase per million mapped reads (FPKM). Because pair-ended sequencing was used, reads at enhancers per gene and normalized reads per enhancer were divided by two to adjust to one read per gene or enhancer. TF-bound sites at enhancers per gene were calculated as the sum of bound sites at all enhancers assigned to a gene. Normalized TF-bound sites per enhancer were calculated by dividing the TF-bound sites at enhancers per gene with the sizes of enhancers. Both the reads and bounds sites at enhancers were calculated as the average reads and bound sites from two biological samples.

**RNA-seq data analysis**. The raw reads (average 20.1 million reads, 3 biological replicates for each treatment in Figs. 4 and 2 biological replicates for each treatment in Fig. 7b *Gata2* deletion analysis) were analyzed and quality checked by FastQC. The reads were aligned to the mm10 reference genome using the Spliced Transcripts Alignment to a Reference (STAR, version 2.4.0.1) software[67]. Reads (FPKM) were assembled into reference transcripts and counted using Cufflinks (version 2.2.1). The average reads from two or three biological samples were assessed using Cuffmerge (version 1.0.0). Reads with FPKM < 30 in resting or stimulated BMMCs were filtered out from further analysis. The differential gene expression between the resting and stimulated samples (pooled) was analyzed by using Cuffdiff (version 2.2.1)[68]. Gene ontology enrichment analysis was performed on genes that responded to IgE receptor crosslinking by using the Database for Annotation, Visualization and Integrated Discovery (DAVID, version 6.8)[69,70]. Genes that were upregulated with log$_2$ ratio of stimulated/untreated >2 in each enriched categories were selected for heatmap representations, which were generated by using Expander software[71] (version 7.2). The dendrogram tree for different expression patterns was calculated using the agglomerative algorithm[72]. The representative RNA-seq IGV tracks showed in the figures were generated from one biological sample, representing two or three biological samples with similar pattens.

**Omni-ATAC-seq data analysis**. Raw sequencing reads (average 118.2 million reads, 2 biological replicates for each treatment) were adapter trimmed by Trimmomatic 0.33 and the quality of sequenced data was analyzed by FastQC. The trimmed reads were aligned to the mm10 reference genome using Bowtie2 with -very-sensitive and -x 2000 parameters. The read alignments were filtered using SAMtools (version 1.7) to remove mitochondrial genome and PCR duplicates. Peaks were identified by MACS2 with the q-value cut-off of 0.05 and the sequencing data was displayed using IGV. Normalized reads per enhancer were calculated by dividing the reads at each enhancer (defined by H3K27ac modification) with the size of enhancers in 1000 bp per unit and were represented as fragments per kilobase per million mapped reads (FPKM). The averages of reads per enhancer were calculated from two biological replicates. The representative ATAC-seq IGV tracks showed in the figures were generated from one biological sample, representing two or three biological samples with similar pattens.

**Calculation of regulatory potential scores for enhancers**. GATA2 ChIP-seq, H3K27ac ChIP-seq or Omni-ATAC-seq BAM files were used to generate peak files using MACS2. The peak files were used for regulatory potential score calculations using Binding and Expression Target Analysis (BETA) minus (Version 1.0.7)[42]. Enhancers located within ±100 kb of TSS of a target gene were assigned to the target gene.

**TF binding motif enrichment analysis**. Multiple Em for Motif Elicitation (MEME)-ChIP was used to identify the TF binding motifs enriched in accessible regions, typical and super-enhancers[73]. To identify the TF binding motifs enriched in the accessible regions located within the regions marked with the H3K27ac modification, BED files of Omni-ATAC-seq data sets were used to intersect with the typical enhancer and super-enhancer regions identified by H3K27ac ChIP-seq using Bedtools intersect (version 2.29.2)[66]. The overlapped sequences were subjected to motif enrichment analysis using MEME-ChIP (version 5.2.0) and the JASPAR database. The number of TF motifs was counted by using the Find Individual Motif Occurrences (FIMO) tool of MEME (FIMO). For analysis of TF binding motifs enriched at the activation-induced accessible regions located within

the activation-induced enhancers, Omni-ATAC-seq or H3K27ac ChIP-seq reads of BMMC samples that were untreated with IgE receptor crosslinking were compared with those of BMMC samples that were treated with IgE receptor crosslinking using the R package DiffBind (version 2.10.0). The regions that had increased H3K27ac ChIP-seq or Omni-ATAC-seq reads after IgE receptor crosslinking were compared using Bedtools intersect. The sequences of the regions that showed both activation-induced H3K27ac ChIP-seq reads and Omni-ATAC-seq reads were used for TF binding motif enrichment analysis using MEME-ChIP.

**Chromatin accessibility enrichment distribution analysis at TSS of the selected regions**. Enrichment scores were calculated by using deepTools 3.3.0[60]. The BigWig files of Omni-ATAC-seq generated from resting BMMCs were used for chromatin accessibility enrichment distribution analysis. The TSS files for the cytokine and chemokine genes were compiled from the TSS files generated by UCSC table browser using the mm10 assembly.

**β-Hexosaminidase release assay**. Mast cell degranulation was measured by the release of β-hexosaminidase. BMMCs cultured from bone marrow cells of inducible Gata2 knockout mice or control mice were treated with 4-OHT for five days and the YFP⁺ cells were FACS-sorted. The sorted cells ($1 \times 10^5$ cells/ml) were sensitized with 1 μg/mL IgE anti-TNP antibody for 24 h and challenged with 100 ng/mL TNP-BSA for 30 minutes. Both cells and supernatants were collected. β-hexosaminidase in supernatants and cell pellets was incubated with an equal volume of 7.5 mM p-nitrophenyl N-acetyl-α-D-glucosaminide (N8759, Sigma-Aldrich), which was dissolved in citric acid (80 mM, pH 4.5) for 2 h at 37 °C. After incubation, the reaction was developed by adding 0.2 M glycine (pH 10.7) and the absorbances at 405 and 490 nm were measured. The percentage of β-hexosaminidase release was calculated as the percentage of β-hexosaminidase in supernatants relative to total β-hexosaminidase in supernatants and cell pellets.

**ELISA measurement of chemokine and cytokine productions**. ELISA was used to measure the amounts of chemokines and cytokines released from BMMCs. BMMCs were sensitized with 1 μg/mL IgE anti-TNP antibody for 24 h and challenged with 100 ng/mL TNP-BSA for 2, 4, 6, 8 and 24 h. Untreated BMMCs were included as a control. The amounts of CCL1 (LS-F290-1, LifeSpan BioSciences, Seattle, WA), CCL2 (LS-F31326-1, LifeSpan BioSciences), CCL3 (LS-F31325-1, LifeSpan BioSciences), CCL4 (LS-F31354-1, LifeSpan BioSciences) and CCL7 (LS-F619-1, LifeSpan BioSciences) protein in the supernatants were measured by ELISA. For cytokine measurement, the sorted Gata2⁻/⁻ or control BMMCs were left untreated or sensitized with 1 μg/mL IgE anti-TNP antibody for 24 h and challenged with 100 ng/mL TNP-BSA overnight. The amounts of IL-6 (LS-F31434-1, LifeSpan BioSciences) and TNFα (LS-F57505-1, LifeSpan BioSciences) protein in the supernatants were measured by ELISA.

**Statistical analysis**. Error bars above the medians represent variants ranging from third quartile to maximum and error bars below the medians represent variants ranging from first quartile to minimum. Two-tailed student's t test was used to determine significant differences between two samples.

**Reporting summary**. Further information on research design is available in the Nature Research Reporting Summary linked to this article.

## Data availability
All raw and processed ATAC-seq, ChIP-seq and RNA-seq data are available in the Gene Expression Omnibus (GEO): "GSE145612". The ATAC-seq data can be found in the GEO database: "GSE145542"; the ChIP-seq data can be found in the GEO database: "GSE145544"; and the RNA-seq data can be found in the GEO database: "GSE145611". The GATA2 ChIP-seq peaks generated from resting BMMCs (Fig. 2) and MITF ChIP-seq data used in Figs. 2 and 7 were downloaded from the GEO database "GSE48086"[38]. All relevant data supporting the key findings of this study are available within the article and its Supplementary Information files or from the corresponding author upon reasonable request. The source data underlying Fig. 1a can be found in Supplementary Data 2; the source data underlying Fig. 1d can be found in Supplementary Data 3; the source data underlying Fig. 3a can be found in Supplementary Data 7; the source data underlying Fig. 4 can be found in Supplementary Data 9; the source data underlying Fig. 5a can be found in Supplementary Data 10; the source data underlying Fig. 6d can be found in Supplementary Data 11; the source data underlying Fig. 7d can be found in Supplementary Data 13; the source data underlying Supplementary Fig. 3 can be found in Supplementary Data 5; the source data underlying Supplementary Fig. 4 can be found in Supplementary Data 6; and the source data underlying Supplementary Fig. 5 can be found in Supplementary Data 8. A reporting summary for this Article is available as a Supplementary Information file.

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

## Acknowledgements

We thank lab members for thoughtful discussions. We are grateful to Drs. James P. Scott-Browne and Philippa Marrack of the Department of Biomedical Research at the National Jewish Health for critical reading of our manuscript; to Ms. Ashley Keating for technical and editorial assistance and Marina Rahmani for technical assistance; to the Cytometry Core for cell sorting; to Animal Care Facility for animal maintenance and technical assistance; to the Gene and Environment Center at the National Jewish Health and the Genomics Core and the Colorado Center for Personalized Medicine at the University of Colorado Anschutz Medical Campus for performing sequencing and assistance for bioinformatic analysis. Supported by grants from the National Institutes of Health R01AI107022 and R01AI135194 (H.H.) and R21 AI138029 (J.H.).

## Author contributions

Y.L., J.G., M.K. and H.H. designed and performed the experiments; Y.L. analyzed the experiments and sequencing data; L.H., T.D., S.M.L., B.P.O. and J.R.H. helped with the data analysis, reviewed and edited the manuscript; Y.L. and H.H. wrote the manuscript; H.H. acquired the funding and supervised the research.

## Competing interests

The authors declare no competing interests.
