## [Peer Review File · Nature Communications]

REVIEWER COMMENTS

Reviewer #1 (Remarks to the Author):

This manuscript reported the regulatory role of transcription factor GATA2 in promoting promotes chromatin accessibility and gene transcription.

The authors performed H3K4me1, H3K27ac ChIP-seq on bone marrow-derived mast cells to identify enhancers and further distinguished super-enhancers from typical enhancers. In addition, ATAC-seq was performed and TF binding motif analysis was done to identify enriched TF binding motifs. A higher frequency of GATA2 binding motifs at the super-enhancers and higher regulatory potential scores were observed.

In the inducible Gata2 knockout bone marrow-derived mast cells, chromatin accessibility at the super-enhancers was found to significantly reduced which is consistent with the significant loss in RNA transcripts.

In vitro stimulation assay was then combined with RNA-seq and ATAC-seq analysis to reveal the increased gene expression and chromatin accessibility after stimulation and activation.

Overall, this comprehensive report was well written with appropriate methodology and statistical analysis. The findings of this manuscript shed new lights into the transcription regulation of mast cells.

The authors stated in the last paragraph "MCs remodel their chromatin landscapes in response to antigen exposure and do not die after antigenic stimulation and degranulation". Were the inducible Gata2 knockout BMMCs functionally responsive? Please comment on the level of mast cell degranulation and β -hexosaminidase release under the different conditions (inducible knockout, resting, activated, etc). Were cytokine levels evaluated?

As it is shown in figure 4c, transcripts of chemokines Ccl1, Ccl2, Ccl3, Ccl4 and Ccl7 increased dramatically in activated mast cells. Did the authors further investigate their findings in the protein level?

Reviewer #2 (Remarks to the Author):

In this study, the authors conducted ChIP-seq with the antibody against H3K27ac and H3K4me1, and Omni-ATAC-seq to find the key transcription factors in resting or stimulated mast cells. They combined the information between super-enhancers and expressions, then determined GATA2 but not MITF promotes chromatin remodeling at super-enhancers of the key MC identity genes and primes chromatin accessibility in inflammatory stimulated MCs, which are the major findings of this study. The authors used various kinds of bioinformatic analyses to enhance the importance of GATA2 and its binding motif. However, whether GATA2 directly binds to key regulatory elements will need further validation during the revision. Major concern, super enhancers are calculated using H3K27ac data in only resting MCs. The authors should examine the changes of both GATA2 and super enhancers at genome-wide level in stimulated MCs because of the lack of direct evidence to conclude the claims in this manuscript.

In addition, the most concern is the conceptual novelty of this manuscript. It is known that SEs are important for the regulation of key genes that determine cell specificity, and that master transcription factor binding to these SEs is important (Cell, 2013; 153: 307-319). And it is also known that after stimulation, the binding of additional transcription factors to primed enhancers is important (Nature, 2013; 503: 487-492).

In addition, many studies have already shown that GATA2 is the master transcription factor for MCs.

Therefore, the most important points are i) whether the data actually supports some of the main

conclusions in the paper ii) given the considerable amount of data and analysis included, the main novel and important findings must be pointed out more clearly and put into context of the already existing knowledge in the field.

Major concerns

1) Genomic and transcriptomic analysis: More information in the main text concerning replicates (source, number) and validation (comparison by Pearson correlation for example)

-Detailed analysis of RNAseq should be shown:

For example, the authors showed the representative genes in IGV using biological duplicate ChIP-seq data in Fig.S1. However, they should show the correlation between duplicate data in genome-wide level. In addition, it has been written that Omni-ATAC-seq, RNA-seq are performed in more than two biologically independent samples in method section. But, they should write in more detail how to calculate the data. The reproducibility has been should be shown, and the authors need to write which data are used? The average, representative, or adding up the data?

2) Fig.5-7

The author should determine and motif analyses about super enhancer regions after stimulation. Recent studies have shown that super-enhancers can change in just a few hours (Mol Cell. 2014 Oct 23;56(2):219-231., EMBO J. 2020 Apr 1;39(7):e103949.). In this paper, the authors should perform the ChIP-seq of GATA2 before and after stimulation of this experimental system because it is the post-stimulation SE that plays an important role in gene regulation after stimulation, whether GATA2 binds in these genomic loci beforehand or after stimulation.

Other concerns

1) Page5, line98-101; We can't easily catch up how to calculate ROSE. The authors used H3K27ac and H3K4me1 data to determine the potential enhancers. But, what data was used to use ROSE has not been described.

2) Page6, line142-144, page10, line 321-322; Supplemental Table 4 only describes the region, but no information on TF binding motif or repression of gene expression.

3) Fig4a; the authors should show the algorithm to make this heatmap.

4) Fig4a; It might be intriguing that the authors discuss about characteristics and functions of the 280 genes that are repressed.

5) Page8, line264; mistake from '21.8' to '21.2'?

6) Page10, line334; Fig.7b?

Reviewer #3 (Remarks to the Author):

This manuscript expands on previous work by the same group, which identified GATA2 and MITF as "lineage-determining transcription factors" for mast cells (MC). Here, they demonstrate that GATA2, but not MITF, regulates both, lineage identity as well as activation status of MC. In the present study, the authors delineate super-enhancers from "typical" enhancers, as well as gene sets conferring lineage identity or antigen responsiveness to MC. Their data demonstrate that GATA2 binding to super enhancers increases chromatin accessibility and transcription of key lineage genes, thereby maintaining MC identity. On the other hand, GATA2 binding to both super and typical enhancers primes activation-responsive genes and thereby contributes to regulating MC activation status. This dual role characterises GATA2 as a "master" transcription factor of MC biology.

This work describes several novel aspects of gene regulation in MC: MC specific enhancers had not previously been described in such detail, and the mechanisms by which GATA2 (and MITF) regulate target genes had been poorly understood, both in the context of lineage identity as well as MC activation. This work identifies a transcriptional mechanism as a putative positive feedback loop enabling rapid cytokine and chemokine responses upon antigenic stimulation of MC.

To my knowledge, such work is novel for the MC community. As illustrated by the first (bulk) transcriptomic data of primary mouse MC being published by the Immunological Genome Consortium only in 2016 (Dwyer et al. Nature Immunology 2016; doi: 10.1038/ni.3445), the MC

field has generally been lacking behind other immunological sub-disciplines with respect to molecular characterisation. This work should, however, also be of interest to the wider Immunology community, considering that many cellular and molecular features are shared between distinct cell types, and long-lived, tissue-resident innate and innate-like lineages like MC and macrophages in particular.

For the latter, eloquent work over the last years has now firmly established that lineage identity and tissue-specificity are established in a step-wise process, whereby a core lineage program is imprinted first that then diversifies under the influence of tissue-specific cues and transcription factors (amongst others, Mass et al. *Science* 2016, doi: 10.1126/science.aaf4238). Once they have taken up tissue residency, macrophage identity thus is primarily determined by the microenvironment rather than developmental origins, at least at homeostasis (Gosselin et al. *Cell* 2014, doi: 10.1016/j.cell.2014.11.023; Lavin et al. *Cell* 2014, doi: 10.1016/j.cell.2014.11.018). Whether similar mechanisms underly MC specification remains to be determined. This is of particular interest since also MC appear to be more heterogenous than previously anticipated. The mechanisms driving this heterogeneity in a tissue-specific manner, as well as the responsible transcription factors, remain completely unknown. Moreover, MC also share similar developmental patterns with macrophages: During development, both are initially established from yolk sac hematopoiesis and gradually get replaced from hematopoietic stem cells (HSC). In the adult steady state, most MC actually self-maintain largely independently from bone marrow (BM) progenitors, however (Gentek et al. *Immunity* 2018, doi: 10.1016/j.immuni.2018.04.025; Li et al. *Immunity* 2018, doi: 10.1016/j.immuni.2018.09.023; Weitzmann et al. *Journal of Investigative Dermatology* 2020, doi: 10.1016/j.jid.2020.03.963). In the light of these developmental patterns, it would be important to delineate the respective roles of GATA2 in specifying the core MC program during ontogeny versus the maintenance of their previously established identity. These considerations also identify the main limitations of the manuscript: Although they are well-established models yielding robust numbers and activation of MC, *in vitro* generated, BM-derived MC and their activation by IgE crosslinking do not represent physiological models of MC biology. While arguably outside the scope of the current study, it would be more desirable to study the role of GATA2 (and additional transcription factors) in primary, tissue-derived MC, as well as MC activated by in more physiological conditions and/or through alternative, non-IgE-mediated pathways that are also increasingly recognized. Furthermore, a major implication of BM-independent self-maintenance of tissue-resident MC is that they are capable of low-grade proliferation even at the mature stage. It has been suggested that macrophages do so through a genetic program reminiscent of embryonic stem cells, which is activated through macrophage-specific enhancers (Soucie et al. *Science* 2016, 10.1126/science.aad5510). Whether a similar gene network is operational in MC, and whether it is regulated by lineage-specific enhancers, has not been explored to date. Seeing that the genes contained in this "self-renewal" network are conserved between embryonic stem cells and differentiated macrophages, it is possible that they also underlie MC maintenance, as also suggested by a very recent report (Weitzmann et al. *Journal of Investigative Dermatology* 2020, doi: 10.1016/j.jid.2020.03.963). Given its key role in maintaining MC identity and priming their activation, MC self-renewal may also be under (partial) control of GATA2.

Overall, this study provides the rationale and framework for important follow-up work. The work appears to have been carried out with sufficient scrutiny for other labs to be able to reproduce both the generation of biological samples, as well as key bioinformatics analyses. Finally, the authors also discuss their data in a comprehensive manner, e.g. where they convincingly argue that diminished Fc ϵ r1g expression in absence of GATA2 cannot not explain all effects on activation induced key genes observed in Gata2 $^{-/-}$ MC.

In summary, despite the limitations outlined above, I congratulate the authors on this body of work, and I recommend this manuscript for publication at Nature Communications following minor revisions.

Please find a list of detailed remarks below. With the exception of an analysis of "self-renewal" genes ((2)), these are largely in-text or in-figure modifications, in light of the quality of the submitted work as well as the current pandemic.

- (1) The authors should acknowledge considerations about in vitro generated, BM-derived versus tissue-derived primary MC (see above) in their discussion.
- (2) They should explore the possibility that like macrophages (Soucie et al. Science 2016, 10.1126/science.aad5510), MC access embryonic stem cell-like "self-renewal" genes via lineage-specific enhancer landscapes by investigating these previously annotated self-renewal genes in their data e.g. for GATA2 binding (sites).
- (3) This reviewer feels that although overall well written, the manuscript suffers from an apparent overuse of acronyms and newly-defined terminology, or combinations thereof. For example, "activation-inducible enhancers (are converted) into activation-induced enhancers" (lines 416-417), could be simplified to e.g. "activation-inducible enhancers are triggered". This applies throughout the manuscript and is particularly true where these are turned into acronyms and especially in figures and legends, e.g. KIDG, NIDG then KAIG and non-KAIG, KCCG and non-KCCG (Figures 3, 5). It may be worthwhile considering a combination of acronyms and more commonly used terminology, such as "key ID genes" instead of KIDG.
- (4) Similarly, please consider shortening/simplifying the title of the manuscript. E.g. "GATA2 regulates mast cell identity and responsiveness to antigenic stimulation by promoting chromatin remodelling at super enhancers"
- (5) For the reasons outlined above with respect to MC ontogeny (i.e. differentiation during development) and differentiation, the manuscript may benefit from a better distinction between GATA2 functions in "development" and maintenance of lineage identity.
- (6) Please comment on why different genetic backgrounds were used for the inducible Gata2 deficient and control strains (C57Bl/6) vs Balb/c and provide evidence that strain background does not impact the data obtained.
- (7) Figure 4a: Are the cut-offs defined somewhere for "highly induced" versus "induced genes"? Please specify in Figure legend.
- (8) Figure 7c: Please specify which cells are analysed by flow cytometry here, i.e. BMMC? How long after differentiation, etc?
- (9) Line 429: Should this read zink finger instead of "zinger"?

Signed:
Rebecca Gentek

We are grateful for the reviewers' comments and suggestions. We think that these comments and suggestions have helped us significantly improve the quality of our work.

We have added additional experiments, including protein data and GATA2 ChIP-seq data. We have also improved figure presentation. In response to the reviewers, we have made the suggested changes. Detailed information regarding revision is included in our point-by-point responses. Changes made to this revised manuscript are marked by Yellow. We hope that these changes will make our revised manuscript acceptable for publication.

Point-by-point responses:

Reviewer #1:

This manuscript reported the regulatory role of transcription factor GATA2 in promoting promotes chromatin accessibility and gene transcription.

The authors performed H3K4me1, H3K27ac ChIP-seq on bone marrow-derived mast cells to identify enhancers and further distinguished super-enhancers from typical enhancers. In addition, ATAC-seq was performed and TF binding motif analysis was done to identify enriched TF binding motifs. A higher frequency of GATA2 binding motifs at the super-enhancers and higher regulatory potential scores were observed.

In the inducible *Gata2* knockout bone marrow-derived mast cells, chromatin accessibility at the super-enhancers was found to significantly reduced which is consistent with the significant loss in RNA transcripts.

In vitro stimulation assay was then combined with RNA-seq and ATAC-seq analysis to reveal the increased gene expression and chromatin accessibility after stimulation and activation.

Overall, this comprehensive report was well written with appropriate methodology and statistical analysis. The findings of this manuscript shed new lights into the transcription regulation of mast cells.

1. The authors stated in the last paragraph “MCs remodel their chromatin landscapes in response to antigen exposure and do not die after antigenic stimulation and degranulation”. Were the inducible *Gata2* knockout BMMCs functionally responsive? Please comment on the level of mast cell degranulation and β -hexosaminidase release under the different conditions (inducible knockout, resting, activated, etc). Were cytokine levels evaluated?

Response: We thank the reviewer for his insightful comments. We examined the function of *Gata2*^{-/-} MCs at the protein level. *Gata2*^{-/-} MCs released similar amounts of β -hexosaminidase to WT MCs under resting conditions but released 74% less β -hexosaminidase compared to WT MCs after IgE receptor crosslinking (Supplementary Fig. 10a). *Gata2*^{-/-} MCs completely failed to make detectable IL-6 protein and only produced 7% TNF- α protein compared to WT BMMCs

in response to IgE receptor crosslinking (Supplementary Fig. 10b). We have added method description (page 20, lines 735 to 758) and text description (page 11-12, lines 402 to 407).

2. As it is shown in figure 4c, transcripts of chemokines Ccl1, Ccl2, Ccl3, Ccl4 and Ccl7 increased dramatically in activated mast cells. Did the authors further investigate their findings in the protein level?

Response: We detect significant increases in these chemokines in resting and activated BMMCs using specific ELISA assays. New data were added to as Supplementary Fig. 7 and we have added text description (page 9, lines 274 to 276) and method description (page 20, lines 747 to 758).

Reviewer #2:

In this study, the authors conducted ChIP-seq with the antibody against H3K27ac and H3K4me1, and Omni-ATAC-seq to find the key transcription factors in resting or stimulated mast cells. They combined the information between super-enhancers and expressions, then determined GATA2 but not MITF promotes chromatin remodeling at super-enhancers of the key MC identity genes and primes chromatin accessibility in inflammatory stimulated MCs, which are the major findings of this study. The authors used various kinds of bioinformatic analyses to enhance the importance of GATA2 and its binding motif. However, whether GATA2 directly binds to key regulatory elements will need further validation during the revision. Major concern, super enhancers are calculated using H3K27ac data in only resting MCs. The authors should examine the changes of both GATA2 and super enhancers at genome-wide level in stimulated MCs because of the lack of direct evidence to conclude the claims in this manuscript.

In addition, the most concern is the conceptual novelty of this manuscript. It is known that SEs are important for the regulation of key genes that determine cell specificity, and that master transcription factor binding to these SEs is important (Cell, 2013; 153: 307-319). And it is also known that after stimulation, the binding of additional transcription factors to primed enhancers is important (Nature, 2013; 503: 487-492).

In addition, many studies have already shown that GATA2 is the master transcription factor for MCs.

Therefore, the most important points are i) whether the data actually supports some of the main conclusions in the paper ii) given the considerable amount of data and analysis included, the main novel and important findings must be pointed out more clearly and put into context of the already existing knowledge in the field.

Response: We have revised the discussion section to emphasize several new findings of our paper. First, we and others have demonstrated that GATA2 is essential for the differentiation of MC progenitor cells into the MC lineage and for maintaining the MC identity once MCs fully committed into the MC lineage. However, it has not been investigated previously how GATA2 regulates enhancers of its target genes in MCs (page 13, lines 413 to 416). Here, we identify

enhancers and super-enhancers that control genes associated with this mechanism. Secondly, although it is known that super-enhancers are often associated with genes that confer cell identities, the mechanisms by which super-enhancers regulate cell identity genes have not been completely delineated. Furthermore, super-enhancers might function differently in different cell types. In MCs, while TFs including GATA2 have been implicated as key contributors to tissue-specific gene expression, this had not been formally shown in the context of super-enhancers that act in MC differentiation. Moreover, the transcriptional control of MC responsiveness to antigenic stimulation had not been investigated previously. Our findings reveal that GATA2 binding is directly associated with high levels of transcription associated with MC identity genes and robust responsiveness to antigenic stimulation (page 13, lines 427 to 434). Finally, our work extends previous findings concerning TF functions in macrophages. It was reported that transcription factor PU.1 primes some genes in macrophages to respond to TLR stimulation. However, the work was conducted prior to the advancement of next generation sequencing and only a few genes were evaluated in these cells. Our proposed model sheds additional light concerning the priming of enhancers that respond to antigenic stimulation. Our work adds new evidence support the LDTF priming model that can be applied to many more genes and additional cell types (page 14, lines 490 to 494).

We have performed GATA2 ChIP-seq under resting and stimulated conditions and these new data strongly support that GATA2 primes enhancers to respond to antigenic stimulation (see our response to the major concern 2) below.

Major concerns

1) Genomic and transcriptomic analysis: More information in the main text concerning replicates (source, number) and validation (comparison by Pearson correlation for example)
-Detailed analysis of RNAseq should be shown:

For example, the authors showed the representative genes in IGV using biological duplicate ChIP-seq data in Fig.S1. However, they should show the correlation between duplicate data in genome-wide level.

In addition, it has been written that Omni-ATAC-seq, RNA-seq are performed in more than two biologically independent samples in method section. But, they should write in more detail how to calculate the data.

The reproducibility should been shown, and the authors need to write which data are used? The average, representative, or adding up the data?

Response: We have provided more information regarding replicates and validation in the main text. For example, we performed H3K4me1, H3K27ac ChIP-seq on two biological replicates of resting bone marrow-derived MCs (BMMCs) (page 5, line 89, 90). We found an average of total 9,517 typical enhancers and 667 super-enhancers found in two replicates (page 5, line 96, 97). Representative tracks from one of the two biological replicates for the typical and super-enhancers are shown in Fig. 1c (page 5, line 105, 106). For RNA-seq data, we have clarified in the result section that three biological replicates were used for RNA-seq analysis (page 8, line 259) and indicated that an average of 1089 genes (calculated from 3 biological replicates) were

upregulated (page 8, line 260). For Omni-ATAC-seq data analysis, we have added text in the result section (page 7, line 213) and added more detailed description in the method section (page 19, lines 699 to 704). We have also added more detailed methods for the calculation of reads and bound sites at enhancers, the average reads and bound sites from two biological replicates were used. These detailed descriptions were added in the method section (page 18, lines 671 to 672, page 19, lines 699 to 702). Detailed RNA-seq analysis, including the complete list of genes in untreated and IgE receptor crosslinking-treated BMMCs and the lists of GO analysis results, are now shown in the Supplementary Table 9 (page 8, line 261, page 9, lines 268, 271, 273, 274 and 284 to 287). Detailed calculation of average reads and detailed description of heatmap generation are now included in the method section (page 19, lines 678 to 681 and 685 to 689).

We analyzed reproducibility of the ChIP-seq and Omni-ATAC-seq data genome-wide by using the deepTools 3.3.059 multiBamSummary and plotCorrelation functions. RNA-seq data reproducibility was analyzed by ggscatter function in R package ggpubr (0.2). We have added text description in the result section (page 5, lines 106 to 107) and a method section to describe how we calculated Pearson correlation coefficients for two biological replicates (page 17, lines 615 to 625). The reproducibility analysis results were added in Supplementary Fig. 1 and 2.

2) Fig.5-7

The author should determine and motif analyses about super enhancer regions after stimulation. Recent studies have shown that super-enhancers can change in just a few hours (Mol Cell. 2014 Oct 23;56(2):219-231., EMBO J. 2020 Apr 1;39(7):e103949.). In this paper, the authors should perform the ChIP-seq of GATA2 before and after stimulation of this experimental system because it is the post-stimulation SE that plays an important role in gene regulation after stimulation, whether GATA2 binds in these genomic loci beforehand or after stimulation.

Response: We have performed GATA2 ChIP-seq on BMMCs under resting and stimulated conditions. The new data were added to Fig. 7b. We did not observe significant changes in GATA2 binding both at the typical or the super-enhancers before and after stimulation (page 11, lines 367-371 and lines 375-379). These data support a model in which GATA2 binds to both typical and super-enhancers in resting MCs before they are stimulated.

Other concerns

1) Page5, line98-101; We can't easily catch up how to calculate ROSE. The authors used H3K27ac and H3K4me1 data to determine the potential enhancers. But, what data was used to use ROSE has not been described.

Response: We have added the description that H3K27ac ChIP-seq data was used for ROSE (page 5, lines 94 to 96).

2) Page6, line142-144, page10, line 321-322; Supplemental Table 4 only describes the region, but no information on TF binding motif or repression of gene expression.

Response: We performed TF binding motif enrichment analysis in these regions (Supplementary Fig. 3) and added the description of these TF binding motifs (page 6, lines 147 to 153, page 10, line 353 to 355). Most of the enriched TF motifs are reported to be associated with transcriptional repressive activities. Potential target genes regulated by accessible regions are now added to Supplementary Table 5. We have discussed potential implications of repressing these genes in activated mast cells (page 15, lines 528 to 533).

3) Fig4a; the authors should show the algorithm to make this heatmap.

Response: We have added the description of the algorithm to make the heatmap in method section (page 19, lines 685 to 689).

4) Fig4a; It might be intriguing that the authors discuss about characteristics and functions of the 280 genes that are repressed.

Response: GO enrichment analysis of the repressed genes were added. Genes that were repressed by IgE receptor crosslinking were found significantly enriched in gene sets involved in cell proliferation, chemotaxis, signal transduction, apoptosis or enriched in genes encoding receptors. The GO enrichment analysis for the repressed genes is included in Supplementary Table 9. We have added text to describe these genes in the result section (page 9, lines 284 to 287) and have discussed these results in the discussion section (page 15, lines 519 to 527)

5) Page8, line264; mistake from '21.8' to '21.2'?

Response: We have changed “21.8” to “21.2” (page 9, line 292).

6) Page10, line334; Fig.7b?

Response: We have changed “Fig. 7a” to “Fig. 7b” (page 11, line 366).

Reviewer #3:

This manuscript expands on previous work by the same group, which identified GATA2 and MITF as “lineage-determining transcription factors” for mast cells (MC). Here, they demonstrate that GATA2, but not MITF, regulates both, lineage identity as well as activation status of MC. In the present study, the authors delineate super-enhancers from “typical” enhancers, as well as gene sets conferring lineage identity or antigen responsiveness to MC. Their data demonstrate that GATA2 binding to super enhancers increases chromatin accessibility and transcription of key lineage genes, thereby maintaining MC identity. On the other hand, GATA2 binding to both super and typical enhancers primes activation-responsive genes and thereby contributes to regulating MC activation status. This dual role characterises GATA2 as a “master” transcription factor of MC biology.

This work describes several novel aspects of gene regulation in MC: MC specific enhancers had not previously been described in such detail, and the mechanisms by which GATA2 (and MITF) regulate target genes had been poorly understood, both in the context of lineage identity as well as MC activation. This work identifies a transcriptional mechanism as a putative positive feedback loop enabling rapid cytokine and chemokine responses upon antigenic stimulation of MC.

o my knowledge, such work is novel for the MC community. As illustrated by the first (bulk) transcriptomic data of primary mouse MC being published by the Immunological Genome Consortium only in 2016 (Dwyer et al. *Nature Immunology* 2016; doi: 10.1038/ni.3445), the MC field has generally been lacking behind other immunological sub-disciplines with respect to molecular characterisation. This work should, however, also be of interest to the wider Immunology community, considering that many cellular and molecular features are shared between distinct cell types, and long-lived, tissue-resident innate and innate-like lineages like MC and macrophages in particular.

For the latter, eloquent work over the last years has now firmly established that lineage identity and tissue-specificity are established in a step-wise process, whereby a core lineage program is imprinted first that then diversifies under the influence of tissue-specific cues and transcription factors (amongst others, Mass et al. *Science* 2016, doi: 10.1126/science.aaf4238). Once they have taken up tissue residency, macrophage identity thus is primarily determined by the microenvironment rather than developmental origins, at least at homeostasis (Gosselin et al. *Cell* 2014, doi: 10.1016/j.cell.2014.11.023; Lavin et al. *Cell* 2014, doi: 10.1016/j.cell.2014.11.018). Whether similar mechanisms underly MC specification remains to be determined. This is of particular interest since also MC appear to be more heterogenous than previously anticipated. The mechanisms driving this heterogeneity in a tissue-specific manner, as well as the responsible transcription factors, remain completely

unknown. Moreover, MC also share similar developmental patterns with macrophages: During development, both are initially established from yolk sac hematopoiesis and gradually get replaced from hematopoietic stem cells (HSC). In the adult steady state, most MC actually self-maintain largely independently from bone marrow (BM) progenitors, however (Gentek et al. *Immunity* 2018, doi: 10.1016/j.immuni.2018.04.025; Li et al. *Immunity* 2018, doi: 10.1016/j.immuni.2018.09.023; Weitzmann et al. *Journal of Investigative Dermatology* 2020, doi: 10.1016/j.jid.2020.03.963). In the light of these developmental patterns, it would be important to delineate the respective roles of GATA2 in specifying the core MC program during ontogeny versus the maintenance of their previously established identity. These considerations also identify the main limitations of the manuscript: Although they are well-established models yielding robust numbers and activation of MC, in vitro generated, BM-derived MC and

their activation by IgE crosslinking do not represent physiological models of MC biology. While arguably outside the scope of the current study, it would be more desirable to study the role of GATA2 (and additional transcription factors) in primary, tissue-derived MC, as well as MC activated by in more physiological conditions and/or through alternative, non-IgE-mediated pathways that are also increasingly recognized. Furthermore, a major implication of BM-

independent self-maintenance of tissue-resident MC is that they are capable of low-grade proliferation even at the mature stage. It has been suggested that macrophages do so through a genetic program reminiscent of embryonic stem cells, which is activated through macrophage-specific enhancers (Soucie et al. Science 2016, 10.1126/science.aad5510). Whether a similar gene network is operational in MC, and whether it is regulated by lineage-specific enhancers, has not been explored to date. Seeing that the genes contained in this

“self-renewal” network are conserved between embryonic stem cells and differentiated macrophages, it is possible that they also underlie MC maintenance, as also suggested by a very recent report (Weitzmann et al. Journal of Investigative Dermatology 2020, doi: 10.1016/j.jid.2020.03.963). Given its key role in maintaining MC identity and priming their activation, MC self-renewal may also be under (partial) control of GATA2.

Overall, this study provides the rationale and framework for important follow-up work. The work appears to have been carried out with sufficient scrutiny for other labs to be able to reproduce both the generation of biological samples, as well as key bioinformatics analyses. Finally, the authors also discuss their data in a comprehensive manner, e.g. where they convincingly argue that diminished *Fcgr1g* expression in absence of GATA2 cannot not explain all effects on activation induced key genes observed in *Gata2*^{-/-} MC.

In summary, despite the limitations outlined above, I congratulate the authors on this body of work, and I recommend this manuscript for publication at Nature Communications following minor revisions.

Please find a list of detailed remarks below. With the exception of an analysis of “self-renewal” genes ((2)), these are largely in-text or in-figure modifications, in light of the quality of the submitted work as well as the current pandemic.

(1) The authors should acknowledge considerations about in vitro generated, BM-derived versus tissue-derived primary MC (see above) in their discussion.

Response: BMDCs represent less mature MCs. Phenotypically, BMDCs resemble mucosal MCs (J Immunol August 1, 1983, 131 (2) 915-922; Nat Rev Immunol. 2014 Jul;14(7):478-94. doi: 10.1038/nri3690). For example, the *Mcpt1* and *Mcpt2* genes were expressed at higher levels in mucosal MCs relative to connective tissue MCs, whereas the *Mcpt5* gene was expressed at higher levels in connective tissue MCs relative to mucosal MCs. In the Immunological Genome Project, Dwyer et al. performed microarray analysis of primary mouse connective tissue MCs isolated from trachea, tongue, esophagus, skin and peritoneum and defined ID genes in these connective tissue MCs. Despite the differences exist between the directly *ex vivo* MCs and *in vitro* cultured BMDCs and the difference exist between the mucosal and connective tissue MCs, our key ID gene list shares 50% of identities with the Dwyer ID gene list (Supplementary Table 4) (page 5, lines 127 to 130, page 6, lines 131 to 136).

(2) They should explore the possibility that like macrophages (Soucie et al. Science 2016, 10.1126/science.aad5510), MC access embryonic stem cell-like “self-renewal” genes via

lineage-specific enhancer landscapes by investigating these previously annotated self-renewal genes in their data e.g. for GATA2 binding (sites).

Response: We thank this reviewer for pointing us to an important area of research. Mast cells are long-live cells. We found that BMMCs expressed some of orthologous the self-renewal genes that have been defined in human embryonic stem cells and macrophages (Supplementary Table 8). We analyzed GATA2 binding sites in these genes and found significant GATA2 binding in the enhancers of these genes (Supplementary Fig. 5). However, deletion of GATA2 did not affect the expression of these potential self-renewal genes (Supplementary Table 8). We added text to describe these findings (page 8, lines 244 to 249).

(3) This reviewer feels that although overall well written, the manuscript suffers from an apparent overuse of acronyms and newly-defined terminology, or combinations thereof. For example, “activation-inducible enhancers (are converted) into activation-induced enhancers” (lines 416-417), could be simplified to e.g. “activation-inducible enhancers are triggered”. This applies throughout the manuscript and is particularly true where these are turned into acronyms and especially in figures and legends, e.g. KIDG, NIDG then KAIG and non-KAIG, KCCG and non-KCCG (Figures 3, 5). It may be worthwhile considering a combination of acronyms and more commonly used terminology, such as “key ID genes” instead of KIDG.

Response: We used the word “triggered” to describe the activation of the inducible enhancers (page 14, lines 457 and 468). We have changed the acronyms used in figures, figure legends, and supplementary tables according to the reviewer’s comment. KIDG was changed to key ID genes, NIDG was changed to Non-ID genes, KAIG was changed to key activation-induced genes, non-KAIG was changed to non-key activation-induced genes, KCCG was changed to key cytokine and chemokine genes, non-KCCG was changed to non-key cytokine and chemokine genes.

(4) Similarly, please consider shortening/simplifying the title of the manuscript. E.g. “GATA2 regulates mast cell identity and responsiveness to antigenic stimulation by promoting chromatin remodelling at super enhancers”

Response: Thank you for the title suggestion. The new title is “GATA2 regulates mast cell identity and responsiveness to antigenic stimulation by promoting chromatin remodeling at super-enhancers”.

(5) For the reasons outlined above with respect to MC ontogeny (i.e. differentiation during development) and differentiation, the manuscript may benefit from a better distinction between GATA2 functions in “development” and maintenance of lineage identity.

Response: We have clarified the role of GATA2 in differentiation of progenitor cells into MC lineage and in maintaining MC identity (page 3, lines 42, 43, 46 to 48 and 52).

(6) Please comment on why different genetic backgrounds were used for the inducible Gata2

deficient and control strains (C57Bl/6) vs Balb/c and provide evidence that strain background does not impact the data obtained.

Response: We have performed Pearson correlation coefficient analysis on RNA-seq gene sets generated from Balb/c and C57BL/6 BMMCs under resting and stimulated conditions. R values of correlation coefficients between the RNA-seq gene sets generated from Balb/c and C57BL/6 BMMCs under resting conditions were 0.86 and 0.90 under stimulated conditions, indicating that the majority of genes expressed under resting and stimulated conditions were very similar (Supplementary Fig. 4). Text description is added (page 7, lines 200 to 209). We presented the comparison between the transcripts of ID genes in resting BMMCs and the key cytokine and chemokine genes under resting and stimulated conditions in Supplementary Table 6.

(7) Figure 4a: Are the cut-offs defined somewhere for “highly induced” versus “induced genes”? Please specify in Figure legend.

Response: We have added our definitions of “highly induced”, “induced”, “unchanged” and “repressed” genes in the figure legend (page 29, lines 986 to 989).

(8) Figure 7c: Please specify which cells are analysed by flow cytometry here, i.e. BMMC? How long after differentiation, etc?

Response: We have added the description of cells in the figure legend (page 30, lines 1018 to 1019). And more detailed description included in the Methods section (page 16, lines 549 to 556).

(9) Line 429: Should this read zink finger instead of “zinger”?

Response: We have changed “zinger” to “zinc finger” (page 14, line 482).

REVIEWERS' COMMENTS

Reviewer #1 (Remarks to the Author):

The original manuscript reported the regulatory role of transcription factor GATA2 in promoting promotes chromatin accessibility and gene transcription.

In the revised manuscript, the authors have added new data as supporting evidence that address the questions and concerns raised in previous comments of the reviewers. The analysis of changes in b-hexosaminidase release, IL-6, TNF- α , and chemokine levels (Ccl1, Ccl2, Ccl3, Ccl4 and Ccl7) has generated consistent results with data obtained with other methods, and hence supports the argument and conclusions the authors drawn.

In the response and the revised text, the authors have well-demonstrated the novelty of their findings over existing knowledge in the field. Overall, this manuscript shed new lights into MC transcription regulation. I would recommend this manuscript for publication at Nature Communications.

Reviewer #2 (Remarks to the Author):

The authors raised all of our concerns with precise point by point answer and major improvements. Notably, the new GATA2 ChIP-seq data in stimulated BMMCs has strengthen this work conclusion. I suggest to accept.

Reviewer #3 (Remarks to the Author):

Having studied the revised manuscript, it is evident that the authors have put great effort in answering the reviewers' requests, both in writing and with providing new data or analyses. I feel that the manuscript now answers these requests sufficiently, and that it benefitted from these revisions. I am therefore happy with the changes the authors have implemented, and recommend the manuscript for publication.

The only minor remaining issue that I would like to point out is that the manuscript should undergo a final round of careful proofreading for spelling mistakes, e.g. line 244: Mast cells are long-live cells, which should be long-lived.

Other than that, I would like to congratulate the authors on a comprehensive body of work that will be valuable for the mast cell and larger immunology communities.

REVIEWERS' COMMENTS

Reviewer #1 (Remarks to the Author):

The original manuscript reported the regulatory role of transcription factor GATA2 in promoting promotes chromatin accessibility and gene transcription.

In the revised manuscript, the authors have added new data as supporting evidence that address the questions and concerns raised in previous comments of the reviewers. The analysis of changes in b-hexosaminidase release, IL-6, TNF- α , and chemokine levels (Ccl1, Ccl2, Ccl3, Ccl4 and Ccl7) has generated consistent results with data obtained with other methods, and hence supports the argument and conclusions the authors drawn.

In the response and the revised text, the authors have well-demonstrated the novelty of their findings over existing knowledge in the field. Overall, this manuscript shed new lights into MC transcription regulation. I would recommend this manuscript for publication at Nature Communications.

Response: We appreciated your positive comments that help us substantially improve our manuscript.

Reviewer #2 (Remarks to the Author):

The authors raised all of our concerns with precise point by point answer and major improvements. Notably, the new GATA2 CHIP-seq data in stimulated BMMCs has strengthen this work conclusion. I suggest to accept.

Response: We thank your insightful comments and suggestion to perform GATA2 CHIP-seq in stimulated mast cells. We also appreciate your suggestion to highlight the new findings from our study.

Reviewer #3 (Remarks to the Author):

Having studied the revised manuscript, it is evident that the authors have put great effort in answering the reviewers' requests, both in writing and with providing new data or analyses. I feel that the manuscript now answers these requests sufficiently, and that it benefitted from these revisions. I am therefore happy with the changes the authors have implemented, and recommend the manuscript for publication.

The only minor remaining issue that I would like to point out is that the manuscript should undergo a final round of careful proofreading for spelling mistakes, e.g. line 244: Mast cells are long-live cells, which should be long-lived.

Other than that, I would like to congratulate the authors on a comprehensive body of work that will be valuable for the mast cell and larger immunology communities.

Response: We thank you for your expert comments and your kind words of congratulations. We believe your comments, suggestions and encouragements have help us improve our manuscript substantially.

We corrected to “long-lived” and we have thoroughly checked the manuscript to make sure there is no spelling mistakes.